# Bayesian Neural Network Priors Revisited

**Vincent Fortuin**[*]
ETH Zürich, Switzerland
fortuin@inf.ethz.ch

**Adrià Garriga-Alonso**[*]
University of Cambridge, United Kingdom
ag919@cam.ac.uk

**Sebastian W. Ober**
University of Cambridge, United Kingdom
swo25@cam.ac.uk

**Florian Wenzel**
Google AI Berlin, Germany
florianwenzel@google.com

**Gunnar Rätsch**
ETH Zürich, Switzerland
raetsch@inf.ethz.ch

**Richard E. Turner**
University of Cambridge, United Kingdom
ret26@eng.cam.ac.uk

**Mark van der Wilk**[†]
Imperial College London, United Kingdom
m.vdwilk@imperial.ac.uk

**Laurence Aitchison**[†]
University of Bristol, United Kingdom
laurence.aitchison@bristol.ac.uk

## Abstract

Isotropic Gaussian priors are the *de facto* standard for modern Bayesian neural network inference. However, it is unclear whether these priors accurately reflect our true beliefs about the weight distributions or give optimal performance. To find better priors, we study summary statistics of neural network weights in networks trained using stochastic gradient descent (SGD). We find that convolutional neural network (CNN) and ResNet weights display strong spatial correlations, while fully connected networks (FCNNs) display heavy-tailed weight distributions. We show that building these observations into priors can lead to improved performance on a variety of image classification datasets. Surprisingly, these priors mitigate the cold posterior effect in FCNNs, but slightly increase the cold posterior effect in ResNets.

## 1 Introduction

In a Bayesian neural network (BNN), we specify a prior $p(w)$ over the neural network parameters, and compute the posterior distribution over parameters conditioned on training data, $p(w|x, y) = p(y|w, x)p(w)/p(y|x)$. This procedure should give considerable advantages for reasoning about predictive uncertainty, which is especially relevant in the small-data setting. Crucially, to perform Bayesian inference, we need to choose a prior that accurately reflects our beliefs about the parameters before seeing any data (Bayes, 1763; Gelman et al., 2013). However, the most common choice of prior for BNN weights is the simplest one: the isotropic Gaussian. Isotropic Gaussians are used across almost all fields of Bayesian deep learning, ranging from variational inference (e.g., Hernández-Lobato & Adams, 2015; Louizos & Welling, 2017; Dusenberry et al., 2020), sampling-based inference (e.g., Neal, 1992; Zhang et al., 2019), and Laplace's method (e.g., Osawa et al., 2019; Immer et al., 2021b), to even infinite networks (e.g., Lee et al., 2017; Garriga-Alonso et al., 2019). It is troubling that no alternatives are usually considered, since better choices likely exist.

Indeed, despite the progress on more accurate and efficient inference procedures, in some settings, the posterior predictive distribution of BNNs using Gaussian priors still leads to worse predictive performance than a baseline obtained by training the network with standard stochastic gradient descent (SGD) (e.g., Zhang et al., 2019; Heek & Kalchbrenner, 2019; Wenzel et al., 2020a). Surprisingly,

---

[*]Equal contribution.
[†]Equal contribution.

these issues can largely be fixed by artificially reducing posterior uncertainty using "cold posteriors" (Wenzel et al., 2020a). The cold posterior is $p(w|x,y)^{\frac{1}{T}}$ for a temperature $0 < T < 1$, where the original Bayes posterior would be obtained by setting $T = 1$ (see Eq. 1). Using cold posteriors can be interpreted as overcounting the data and, hence, deviating from the Bayesian paradigm. This should not happen if the prior and likelihood accurately reflect our beliefs. Assuming inference is working correctly, the Bayesian solution, $T = 1$, really should be optimal (Gelman et al., 2013). Hence, it raises the possibility that either the prior (Wenzel et al., 2020a) or likelihood (Aitchison, 2020b) (or both) are misspecified.

In this work, we study empirically whether isotropic Gaussian priors are indeed suboptimal for BNNs and whether this can explain the cold posterior effect. We analyze the performance of different BNN priors for different network architectures and compare them to the empirical weight distributions of standard SGD-trained neural networks. We conclude that correlated Gaussian priors are better in ResNets, while uncorrelated heavy-tailed priors are better in fully connected neural networks (FCNNs). Thus, we would recommend these choices instead of the widely-used isotropic Gaussian priors. While these priors eliminate the cold posterior effect in FCNNs, they slightly increase the cold posterior effect in ResNets. This provides evidence that the cold posterior effect arises due to a misspecification of the prior (Wenzel et al., 2020a) in FCNNs. In ResNets, it is difficult to draw any strong conclusions about the cold posterior effect from our results. Our observations are compatible with the hypothesis that the cold posterior effect arises in large-scale image models due to a misspecified likelihood (Aitchison, 2020b) or due to data augmentation (Izmailov et al., 2020), but there could of course be a prior that we did not consider that improves performance and eliminates the cold posterior effect. We make our library available on Github[1], inviting other researchers to join us in studying the role of priors in BNNs using state-of-the-art inference.

## 1.1 CONTRIBUTIONS

Our main contributions are:

- An analysis of the empirical weight distributions of SGD-trained neural networks with different architectures, suggesting that FCNNs learn heavy-tailed weight distributions (Sec. 3.1), while CNN and ResNet weight distributions show significant spatial correlations (Sec. 3.2).

- Experiments in Bayesian FCNNs showing that heavy-tailed priors give better classification performance than the widely-used Gaussian priors (Sec. 4.2).

- Experiments in Bayesian ResNets showing that spatially correlated Gaussian priors give better classification performance than isotropic priors (Sec. 4.3).

- Experiments showing that the cold posterior effect can be reduced by choosing better, heavy-tailed priors in FCNNs, while the cold posterior is slightly increased when using better, spatially correlated priors in ResNets (Sec. 4).

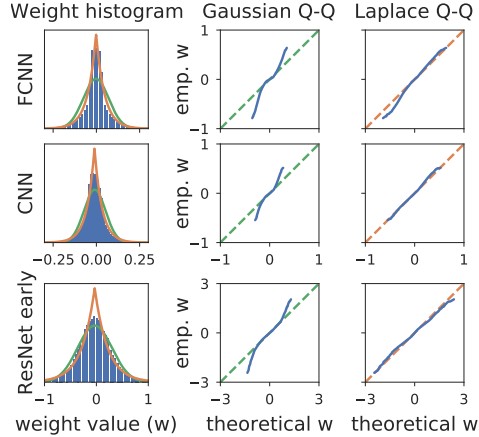

Figure 1: Empirical marginal weight distributions of a layer of FCNNs and CNNs trained with SGD on MNIST, and an early layer of several ResNets trained on CIFAR-10. We show weight histograms (left) and quantile-quantile (Q-Q) plots with different distributions (right). The empirical weights are clearly heavier-tailed than a Gaussian (green line), and better fit by a Laplace (orange line).

---

[1]`https://github.com/ratschlab/bnn_priors`. MIT licensed.

## 2 BACKGROUND: THE COLD POSTERIOR EFFECT

When performing inference in Bayesian models, we can temper the posterior by a positive temperature $T$, giving

$$\log p(w|x,y)^{\frac{1}{T}} = \frac{1}{T}[\log p(y|w,x) + \log p(w)] + Z(T) \tag{1}$$

for neural network weights $w$, inputs $x$ regression targets or class-labels $y$, prior $p(w)$, likelihood $p(y|w,x)$, and a normalizing constant $Z(T)$. Setting $T = 1$ yields the standard Bayesian posterior. The temperature parameter can be easily handled when simulating Langevin dynamics, as used in molecular dynamics and MCMC (Leimkuhler & Matthews, 2012).

In their recent work, Wenzel et al. (2020a) have drawn attention to the fact that cooling the posterior in BNNs (i.e., setting $T < 1$), often improves performance. Testing different hypotheses for potential problems with the inference, likelihood, and prior, they conclude that the BNN priors (which were Gaussian in their experiments) are misspecified—at least when used in conjuction with standard neural network architectures on standard benchmark tasks—which could be one of the main causes of the cold posterior effect (c.f., Germain et al., 2016; van der Wilk et al., 2018). Reversing this argument, we can hypothesize that choosing better priors for BNNs may lead to a less pronounced cold posterior effect, which we can use to evaluate different candidate priors.

## 3 EMPIRICAL ANALYSIS OF NEURAL NETWORK WEIGHTS

As we have discussed, standard Gaussian priors may not be the optimal choice for modern BNN architectures. But how can we find more suitable priors? Since it is hard to directly formulate reasonable prior beliefs about neural network weights, we turn to an empirical approach. We trained fully connected neural networks (FCNNs), convolutional neural networks (CNNs), and ResNets with SGD on various image classification tasks to obtain an approximation of the empirical distribution of the fitted weights, that is, the distribution of the *maximum a posteriori* (MAP) solutions reached by SGD. If the distributions over SGD-fitted weights differ strongly from the usual isotropic Gaussian prior, that provides evidence that those features should be incorporated into the prior. Hence, we can use our insights by inspecting the empirical weight distribution to propose better-suited priors.

Formally, this procedure can be viewed as approximate human-in-the-loop expectation maximization (EM). In particular, in expectation maximization, we alternate expectation (E) and maximization (M) steps. In the expectation (E) step, we infer the posterior $p(w|x,y,\theta_{t-1})$ over the weights, $w$, given the parameters of the prior from the previous step, $\theta_{t-1}$. In our case, we approximately infer the weights using SGD. Then, in the maximization step, we compute new prior parameters $\theta_t$, by sampling weights $w$ from the posterior computed in the E step, and maximizing the joint probability of sampled weights and data. As $y$ is independent of the prior parameters if the weights are known, the M-step reduces to fitting a prior distribution to the weights sampled from the posterior, that is,

$$\mathcal{L}_t(\theta) = \mathbb{E}_{p(w|x,y,\theta_{t-1})}[\log p(y|x,w) + \log p(w|\theta)]$$
$$= \mathbb{E}_{p(w|x,y,\theta_{t-1})}[\log p(w|\theta)] + \text{const} \tag{2}$$
$$\theta_t = \arg\max \mathcal{L}_t(\theta) . \tag{3}$$

Intuitively, this procedure allows the prior (and therefore the posterior) to assign more probability mass to the SGD solutions, which are known to work well in practice. This is also related to ideas from empirical Bayes (Robbins, 1992), where the (few) hyperparameters of the prior are fit to the data, and to recent ideas in PAC-Bayesian theory, where data-dependent priors have been shown to improve generalization guarantees over data-independent ones (Rivasplata et al., 2020; Dziugaite et al., 2021). While such approaches introduce a certain risk of overfitting (Ober et al., 2021), we would argue that standard BNNs are typically thought to be underfitting (Neal, 1996; Wenzel et al., 2020a; Dusenberry et al., 2020) and that we do not directly fit the prior parameters, but merely draw inspiration for the choice of prior family from the qualitative shape of the empirical weight distributions.

We begin by considering whether the weights of FCNNs and CNNs are heavy-tailed, and move on to look at correlational structure in the weights of CNNs and ResNets. Note that in the exploratory experiments here, we used SGD to perform MAP inference with a uniform prior (that is, maximum likelihood fitting). This avoids any prior assumptions obscuring interesting patterns in the inferred

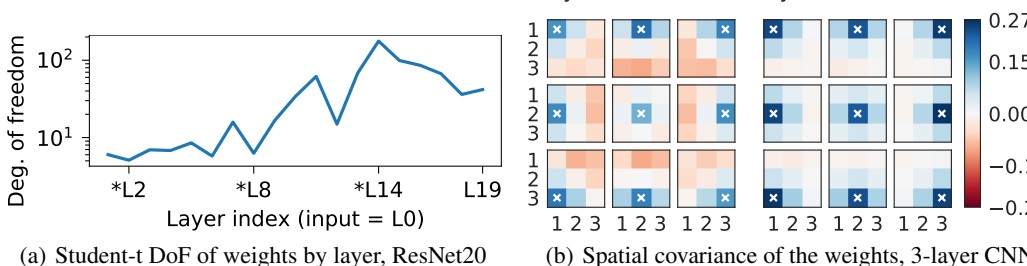

(a) Student-t DoF of weights by layer, ResNet20    (b) Spatial covariance of the weights, 3-layer CNN

Figure 2: **(a)** Degrees of freedom for Student-t distributions fitted to the weights of a ResNet20 trained on CIFAR-10. The degrees of freedom get larger in deeper layers, implying that the weight distributions become less heavy-tailed and more similar to Gaussians. The layers marked with asterisks (*) are the first layers of their respective ResNet blocks. **(b)** Spatial covariance of the weights within CNN filters for a three-hidden layer network trained on MNIST, normalized by the number of channels. The weights correlate strongly with neighboring pixels, and anti-correlate (layer 1) or do not correlate (layer 2) with distant ones. Each delineated square shows the covariances of a filter location (marked with ×) with all other locations.

weights. These patterns inspired our choice of priors, and we then evaluated these priors in BNNs, showing that they improved classification performance (see Sec. 4).

### 3.1 FCNN WEIGHTS ARE HEAVY-TAILED

We trained an FCNN (Fig. 1, top) and a CNN (Fig. 1, middle) on MNIST (LeCun et al., 1998). The FCNN is a three layer network with 100 hidden units per layer and ReLU nonlinearities. The CNN is a three layer network, with two convolutional layers and one fully connected layer. The convolutional layers have 64 channels and use $3 \times 3$ convolutions, followed by $2 \times 2$ max-pooling layers. All layers use ReLU nonlinearities. Networks were trained with SGD for 450 epochs using a learning rate schedule of 0.05, 0.005, and 0.0005 for 150 epochs each. We can see in Figure 1 that the weight values of the FCNNs and CNNs follow a more heavy-tailed distribution than a Gaussian, with the tails being reasonably well approximated by a Laplace distribution. This suggests that "true" BNN priors might be more heavy-tailed than isotropic Gaussians.

Next, we did a similar analysis for a ResNet20 trained on CIFAR-10 (Krizhevsky, 2009) (Fig. 1, bottom). Since this network had many layers, we quantified the degree of heavy-tailedness by fitting the degrees of freedom parameter $\nu$ of a Student-t distribution. For $\nu \to \infty$, the Student-t becomes Gaussian, so large values of $\nu$ indicate that the weights are approximately Gaussian, whereas smaller values indicate heavy-tailed behavior (see Sec. 4.1). We found that at lower layers, $\nu$ was small, so the weights were somewhat heavy-tailed, whereas at higher layers, $\nu$ became much larger, so the weights were approximately Gaussian (Fig. 2a).

These results are perhaps expected if we assume that the filters have (using neuroscience terminology) "localized receptive fields", like those in Olshausen & Field (1997). Such filters contain a large number of near-zero weights outside the receptive field, with a number of very large weights inside the receptive field (Sahani & Linden, 2003; Smyth et al., 2003), and thus will follow a heavy-tailed distribution. As we get into the deeper layers of the networks, receptive fields are expected to become larger, so this effect may be less relevant.

### 3.2 CNN WEIGHTS ARE SPATIALLY CORRELATED

In the second part of our empirical inspection of fitted weight distributions, we looked at spatial correlations in CNN filters. In particular, we considered 9-dimensional vectors formed by the $3 \times 3$ filters for every input and output channel. We studied our three-layer network trained on MNIST and found strong correlations between nearby pixels, and lesser (layer 2) or even negative (layer 1) correlations at more distant pixels (Fig. 2b). We found similar spatial correlations in a ResNet20 trained on CIFAR-10, across all layers, with correlation strength increasing as we move to later layers

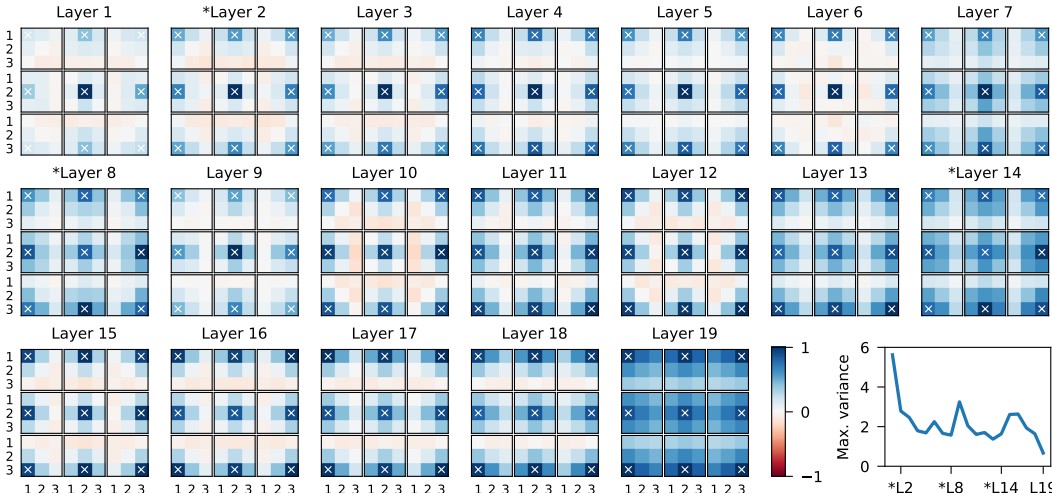

Figure 3: Spatial covariances for the convolutional weights of the layers of a ResNet-20, normalized by the maximum variance for each layer, which is shown on the bottom right. We trained the network with SGD on CIFAR-10 with data augmentation (10 times). Layer 1 is the closest to the input. The first layer of every ResNet block is marked with an asterisk (*). We see that there are significant covariances in all layers, but that their strength increases for later layers.

(Fig. 3). We found by far the strongest evidence of correlations spatially, that is, between weights within the same convolutional filter. This could potentially be due to the smoothness and translation equivariance properties of natural images (Simoncelli, 2009). However, we also found some evidence for spatial correlations in the input layer of an FCNN (Fig. A.1 in the appendix), but no evidence for correlations between the channels of a convolutional layer (Fig. A.5 in the appendix). Note though that this methodology cannot find structured correlations between channels, except at the input and output. This is because NN functions are invariant to permutations of channels (Sussmann, 1992; MacKay, 1992; Bishop et al., 1995; Aitchison, 2020a; Aitchison et al., 2020).

These findings suggest that better priors could be designed by explicitly taking this correlation structure into account. We hypothesize that multivariate distributions with non-diagonal covariance matrices could be good candidates for convolutional layer priors, especially when the covariances are large for neighboring pixels within the convolutional filters (see Sec. 4.3).

Additional evidence for the usefulness of correlated weights comes from the theory of infinitely wide CNNs and ResNets. Novak et al. (2019) noticed that the effect of weight-sharing disappears when infinite filters are used with isotropic priors. More recently, Garriga-Alonso & van der Wilk (2021) showed that this effect can be avoided by using spatially correlated priors, leading to improved performance. Our experiments investigate whether this prior is also useful in the finite-width case.

## 4 EMPIRICAL STUDY OF BAYESIAN NEURAL NETWORK PRIORS

We performed experiments on MNIST and on CIFAR-10. We compare Bayesian FCNNs, CNNs, and ResNets on these tasks. For the BNN inference, we used Stochastic Gradient Markov Chain Monte Carlo (SG-MCMC), in order to scale to large training datasets. To obtain posterior samples that are close to the true posterior, we used an inference method that builds on the inference approach used in Wenzel et al. (2020a), which has been shown to produce high-quality samples. In particular, we combined the gradient-guided Monte Carlo (GG-MC) scheme from Garriga-Alonso & Fortuin (2021) with the cyclical learning rate schedule from Zhang et al. (2019) and the preconditioning and convergence diagnostics from Wenzel et al. (2020a). We ran each chain for 60 cycles of 45 epochs each, taking one sample at the end of each of the last five epochs of each cycle, thus yielding 300 samples after 2,700 epochs, out of which we discarded the first 50 samples as a burn-in. Per temperature setting, dataset, model, and prior, we ran five such chains as replicates. Additional experimental results can be found in Appendix A, details about the evaluation metrics in Appendix B,

about the priors in Appendix C, and about the implementation in Appendix D. In the figures, we generally include an SGD baseline for the predictive error, where it is often competitive with some of the priors. For the likelihood, calibration, and OOD detection, the SGD baselines were out of the plotting range and are therefore not shown. For completeness, we show them in Appendix A.4. We show results for higher temperatures ($T > 1$) in Appendix A.6, for different prior variances in Appendix A.7, and for different network architectures in Appendix A.8. Moreover, while we focus on image classification tasks in this section, we provide results on UCI regression tasks in Appendix A.9. We also show inference diagnostics highlighting the accuracy of our MCMC sampling in Appendix A.10. Finally, we replicate our experiments on ResNets and CIFAR-10 for mean-field variational inference (Blundell et al., 2015) in Appendix A.11.

## 4.1 PRIORS UNDER CONSIDERATION

We contrast the widely used isotropic Gaussian priors with heavy-tailed distributions, including the Laplace and Student-t distributions, and with correlated Gaussian priors. We chose these distributions based on our observations of the empirical weight distributions of SGD-trained networks (see Sec. 3) and for their ease of implementation and optimization. Further details on the distributions and their density functions can be found in Appendix C.

The **isotropic Gaussian** distribution (Gauss, 1809) is the *de-facto* standard for BNN priors in recent work (e.g., Hernández-Lobato & Adams, 2015; Louizos & Welling, 2017; Dusenberry et al., 2020; Wenzel et al., 2020a; Neal, 1992; Zhang et al., 2019; Osawa et al., 2019; Immer et al., 2021b; Lee et al., 2017; Garriga-Alonso et al., 2019). However, its tails are relatively light compared to some of the other distributions that we will consider and compared to the empirical weight distributions described above. The **Laplace** distribution (Laplace, 1774), for instance, has heavier tails than the Gaussian. It is often used in the context of (frequentist) *lasso* regression (Tibshirani, 1996). Similarly, the **Student-t** distribution is also heavy-tailed. Moreover, it can be seen as a Gaussian scale-mixture, where the scales are inverse-Gamma distributed (Helmert, 1875; Lüroth, 1876).

For our correlated Bayesian CNN priors, we use **multivariate Gaussian** priors and define the covariance $\Sigma$ to be block-diagonal, such that the covariance between weights in different filters is 0 and between weights in the same filter is given by a Matérn kernel ($\nu = 1/2$) on the pixel distances. Formally, for the weights $w_{i,j}$ and $w_{i',j'}$ in filters $i$ and $i'$ and for pixels $j$ and $j'$, the covariance is

$$\text{cov}(w_{i,j}, w_{i',j'}) = \begin{cases} \sigma^2 \exp\left(\frac{-d(j,j')}{\lambda}\right) & \text{if } i = i' \\ 0 & \text{otherwise} \end{cases}, \tag{4}$$

where $d(\cdot, \cdot)$ is the Euclidean distance between pixel positions and we set $\sigma = \lambda = 1$. This kernel was chosen to capture the decay with distance of spatial correlations (Fig. 3).

## 4.2 BAYESIAN FCNN PERFORMANCE WITH DIFFERENT PRIORS

Following our observations from the empirical weight distributions (Sec. 3.1), we hypothesized that heavy-tailed priors should work better than Gaussian priors for Bayesian FCNNs. We tested this hypothesis by performing BNN inference with the same network architecture as in Sec. 3, using different priors. We report the predictive error and log likelihood on the MNIST test set. We follow Ovadia et al. (2019) in reporting the calibration of the uncertainty estimates on rotated MNIST digits and the out-of-distribution (OOD) detection accuracy on FashionMNIST (Xiao et al., 2017). For more details about our evaluation metrics, see Appendix B.

We observe that the heavy-tailed priors indeed outperform the Gaussian prior in terms of test error and test NLL in all cases, except for the Student-t distribution on MNIST at low temperatures (Fig. 4). That said, calibration and OOD metrics are less clear, with heavy-tailed priors giving worse calibration and roughly similar OOD detection on MNIST and better calibration but worse OOD detection on FashionMNIST. Despite the unclear results on calibration and OOD detection, the error and NLL performance improvement for heavy-tailed priors at $T = 1$ is considerable, and suggests that Gaussian priors over the weights of FCNNs induce poor priors in the function space and inhibit the posterior from assigning probability mass to high-likelihood solutions, such as the SGD solutions analyzed above (Sec. 3). Finally, the cold posterior effect is removed—or even inverted—when using heavy-tailed priors, which supports the hypothesis that it is caused by prior misspecification in

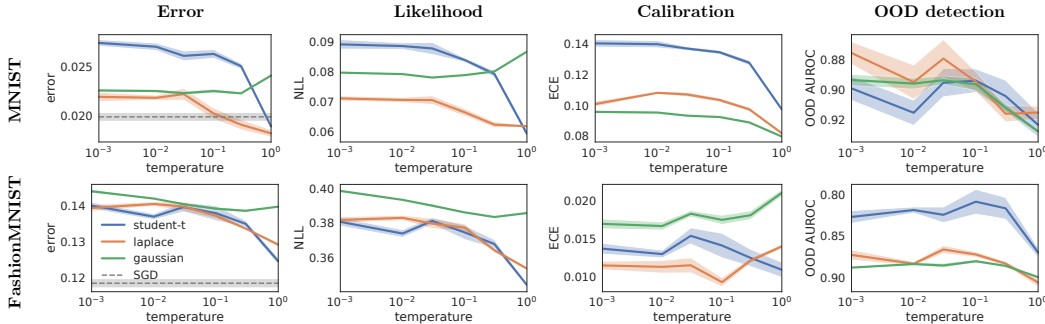

Figure 4: Performances of fully connected BNNs with different priors on MNIST and FashionMNIST (see Sec. 4.2). The heavy-tailed priors generally perform better, especially at higher temperatures, and lead to a less pronounced cold posterior effect. Note the reversed y-axis for OOD detection on the right to ensure that lower values are better in all plots. Shaded regions represent one standard error.

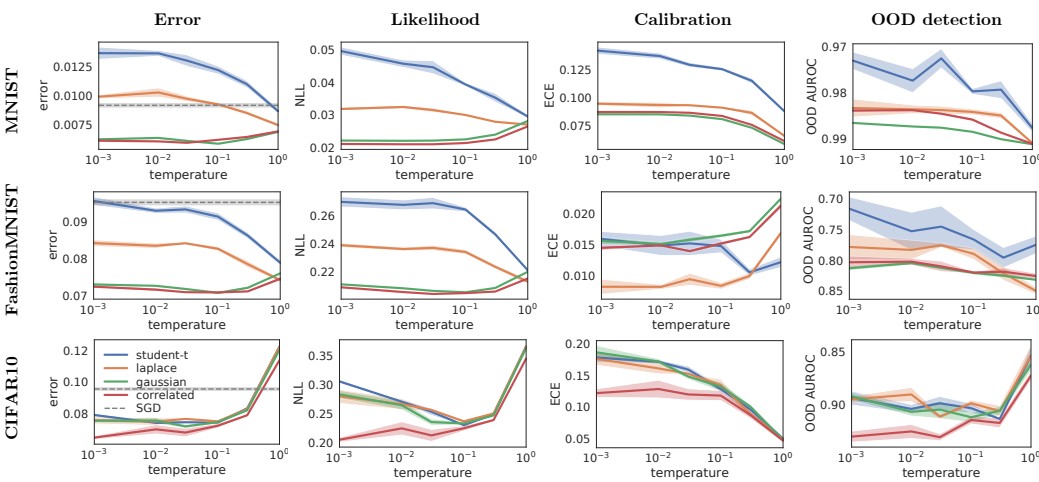

Figure 5: Performances of convolutional BNNs with different priors on MNIST, FashionMNIST, and CIFAR-10 (see Sec. 4.3). The (Fashion)MNIST experiments used CNNs, while the CIFAR-10 experiments used ResNet20. The correlated prior generally performs better than the isotropic ones, but still exhibits a cold posterior effect, while the heavy-tailed priors reduce the cold posterior effect, but yield a worse performance. Note the reversed y-axis for OOD detection on the right to ensure that lower values are better in all plots. Shaded regions represent one standard error.

FCNNs. Note that the cold posterior effect is typically observed in terms of performance metrics like error and NLL, and not calibration and OOD detection performance (Wenzel et al., 2020a). As such, even with Gaussian priors, we do not necessarily expect calibration and OOD detection to exhibit a cold posterior effect. Indeed, only calibration for FashionMNIST exhibits a cold posterior effect, with calibration for MNIST and all OOD detection results exhibiting an inverted cold posterior effect. Notably, we see in Appendix A.5 and Appendix A.7 that these observations generalize to different activation functions and prior variances and in Appendix A.6 that warm posteriors ($T > 1$) deteriorate the performance for all considered priors, such that for the heavy-tailed priors, $T \approx 1$ is indeed ideal.

## 4.3 BAYESIAN CNN AND RESNET PERFORMANCE WITH DIFFERENT PRIORS

We repeated the same experiment for Bayesian CNNs on MNIST and FashionMNIST (Fig. 5, first two rows). Given our observations about SGD-trained weights (Sec. 3.1), we might again expect heavy-tailed priors to outperform Gaussian priors. However, this is not the case: the Gaussian and correlated Gaussian priors perform better in almost all cases, with the exception of calibration for

FashionMNIST. Interestingly, the performance of different methods tends to be very similar at $T = 1$, and to diverge for lower temperatures, with performance improving for Gaussian and correlated Gaussian priors (indicating a cold posterior effect), and worsening for heavy-tailed priors, indicating no cold posterior effect.

Our analysis of SGD-trained weights (Sec. 3.2) also suggested that introducing spatial correlations in the prior (Sec. 4.1) might help. We observe that introducing correlations indeed improves performance compared to the isotropic Gaussian prior (Fig. 5). Notably, the performance improvement is small for CNNs trained on MNIST and FashionMNIST, and for ResNets trained on CIFAR-10 at higher temperatures, but more considerable for ResNets at lower temperatures. As such, correlated priors actually increase the magnitude of the cold posterior effect in ResNets trained on CIFAR-10. This might be because ResNets trained at very low temperatures on CIFAR-10 have a tendency to overfit, and imposing the prior helps to mitigate this overfitting. To support this hypothesis, we indeed see that correlated priors considerably improve over all other methods in terms of calibration and OOD detection at low temperatures for ResNets trained on CIFAR-10.

To reiterate a point raised in Sec. 4.2, the original cold posterior paper (Wenzel et al., 2020a) considered only predictive performance (error and likelihood), and not other measures of uncertainty such as calibration and OOD detection. Indeed, we see different effects of temperature on these measures, with calibration improving for FashionMNIST at lower temperatures, but worsening for MNIST and CIFAR-10. At the same time, we see performance at OOD detection worsen at lower temperatures in the smaller CNN model trained on MNIST and FashionMNIST, but increase at lower temperatures in the ResNet trained on CIFAR-10. These results are consistent with other observations that measures of uncertainty do not necessarily correlate with predictive performance (Ovadia et al., 2019; Izmailov et al., 2021), and indicate that the cold posterior effect is a complex phenomenon that demands careful future investigation. Again, we see in Appendix A.5 and Appendix A.7 that these observations generalize to different activation functions and prior variances and in Appendix A.6 that warm posteriors ($T > 1$) deteriorate the performance for all considered priors.

In practice, models on this dataset are often trained using data augmentation (as is our model in Fig. 5). While this does indeed improve the performance (Fig. A.11 in the appendix), it also strengthens the cold posterior effect. When we do not use data augmentation, the cold posterior effect (at least between $T = 1$ and lower temperatures) is almost entirely eliminated (see Fig. A.11 in the appendix and Wenzel et al., 2020a; Izmailov et al., 2021). This observation raises the question of why data augmentation drives the cold posterior effect. Given that data augmentation adds terms to the likelihood while leaving the prior unchanged, we could expect that the problem is in the likelihood, as was recently argued by Aitchison (2020b). On the other hand, van der Wilk et al. (2018) argued that treating synthetic augmented data as extra datapoints for the purposes of the likelihood is incorrect from a Bayesian point of view. Instead, they express data augmentation in the prior, by constraining the classification functions to be invariant to certain transformations. More investigation is hence needed into how data augmentation and the cold posterior effect relate.

## 5 RELATED WORK

**Empirical analysis of weight distributions.** There is some history in neuroscience of analysing the statistics of data to inform inductive priors for learning algorithms, especially when it comes to vision (Simoncelli, 2009). For instance, it has been noted that correlations help in modeling natural images (Srivastava et al., 2003), as well as sparsity in the parameters (Smyth et al., 2003; Sahani & Linden, 2003). In the context of machine learning, the empirical weight distributions of standard neural networks have also been studied before (Bellido & Fiesler, 1993; Go & Lee, 1999), including the insight that SGD can produce heavy-tailed weights (Gurbuzbalaban & Simsekli, 2020), but these works have not systematically compared different architectures and did not use their insights to inform Bayesian prior choices.

**BNNs in practice.** Since the inception of Bayesian neural networks, scholars have thought about choosing good priors for them, including hierarchical (MacKay, 1992) and heavy-tailed ones (Neal, 1996). In the context of infinite-width limits of such networks (Lee et al., 2017; Matthews et al., 2018; Garriga-Alonso et al., 2019; Yang, 2019; Tsuchida et al., 2019) it has also been shown that networks with very heavy-tailed (i.e., infinite variance) priors have different properties from finite-variance priors (Neal, 1996; Peluchetti et al., 2020). However, most modern applications of BNNs still relied

on simple Gaussian priors. Although a few different priors have been proposed for BNNs, these were mostly designed for specific tasks (Atanov et al., 2018; Ghosh & Doshi-Velez, 2017; Overweg et al., 2019; Nalisnick, 2018; Cui et al., 2020; Hafner et al., 2020) or relied heavily on non-standard inference methods (Sun et al., 2019; Ma et al., 2019; Karaletsos & Bui, 2020; Pearce et al., 2020). Moreover, while many interesting distributions have been proposed as variational posteriors for BNNs (Louizos & Welling, 2017; Swiatkowski et al., 2020; Dusenberry et al., 2020; Ober & Aitchison, 2020; Aitchison et al., 2020), these approaches have still used Gaussian priors. Others use a non-Gaussian prior, but approximate the posterior with a diagonal Gaussian (Blundell et al., 2015; Ghosh & Doshi-Velez, 2017; Nalisnick et al., 2015), somewhat limiting the prior's effect. Another BNN posterior approximation is dropout (Gal & Ghahramani, 2016; Kingma et al., 2015), which is often poorly calibrated (Foong et al., 2019), but can also be seen to induce a scale-mixture prior, similar to our heavy-tailed priors (Molchanov et al., 2017).

**BNN priors.** Finally, previous work has investigated the performance of neural network priors chosen without reference to the empirical distributions of SGD-trained networks (Blundell et al., 2015; Ghosh & Doshi-Velez, 2017; Wu et al., 2018; Atanov et al., 2018; Nalisnick, 2018; Overweg et al., 2019; Farquhar et al., 2019; Cui et al., 2020; Rothfuss et al., 2020; Hafner et al., 2020; Matsubara et al., 2020; Tran et al., 2020; Ober & Aitchison, 2020; Garriga-Alonso & van der Wilk, 2021; Fortuin, 2021; Immer et al., 2021a). While these priors might in certain circumstances offer performance improvements, they did not offer a recipe for finding potentially valuable features to incorporate into the weight priors. In contrast, we offer such a recipe by examining the distribution of weights trained under a uniform prior with SGD. Importantly, unlike prior work, we use SG-MCMC with carefully evaluated convergence metrics and systematically address the cold posterior effect.

Contemporaneous work[2] (Izmailov et al., 2021) compared gold-standard HMC inference with the more practical cyclical SG-MCMC used in our work. They confirmed that cyclical SG-MCMC methods indeed have high-fidelity to the true posterior, and interestingly show that heavy-tailed priors offer slight performance improvements for language modeling tasks (though they do not assess the interaction of the cold posterior effect with these priors).

## 6 CONCLUSION

We consider empirical weight distributions in non-Bayesian networks trained using SGD, finding that FCNNs displayed heavy-tailed weight distributions, and CNNs and ResNets displayed spatial correlations in the convolutional filters. We therefore tested the performance of these priors and their interaction with the cold posterior effect. Indeed, we found that these priors improved performance, but their impact on the cold posterior effect was more complex, with heavy-tailed priors in FCNNs eliminating the cold posterior effect, correlated priors in CNNs trained on MNIST and FashionMNIST leaving the cold posterior largely unchanged, and correlated priors in ResNets trained on CIFAR-10 actually increasing the cold posterior effect, as they yield much larger performance improvements at lower temperatures. Importantly though, we do not expect there to be one "universal" prior that improves performance in all architectures and all tasks. The best prior is almost certain to be highly task- and architecture-dependent, and indeed we found that heavy-tailed priors offer little or no benefits for regression on UCI datasets (Sec. A.9).

Thus, we can conclude that isotropic Gaussian priors are often non-optimal, and that it is worth exploring other priors more generally (as always though, the correct prior will heavily depend on the architecture and dataset). However, it is difficult to come to any strong conclusions regarding the origin of the cold posterior effect. At least in FCNNs, it does indeed appear that a misspecified prior can cause the cold posterior effect. However, in perhaps more relevant large-scale image models, we found that better (correlated) priors actually increase the cold posterior effect, which is consistent with other hypotheses, such as a misspecified likelihood (Aitchison, 2020b), though of course we cannot rule out that there is a better prior that eliminates the cold posterior effect that we did not consider. We hope that our PyTorch library for BNN inference with different priors will catalyze future research efforts in this area and will also be useful on real-world tasks.

---

[2]released on arXiv two months after ours

ACKNOWLEDGMENTS

VF was supported by a PhD fellowship from the Swiss Data Science Center. AGA was supported by a UK Engineering and Physical Sciences Research Council studentship [1950008]. We thank Alexander Immer, Andrew Foong, David Burt, Seth Nabarro, and Kevin Roth for helpful discussions and the anonymous reviewers for valuable feedback. We also thank Edwin Thompson Jaynes for constant inspiration.

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

## A  ADDITIONAL EXPERIMENTAL RESULTS

### A.1  COVARIANCE MATRICES OF FCNN, CNN AND RESNET

Here we report the full covariance matrices for the layers that were analyzed above (Sec. 3.2). We display the covariances for the FCNN for layer 1 (Fig. A.1), layer 2 (Fig. A.2) and layer 3 (Fig. A.3). The only discernable structure is in the first layer, presumably because the weights from neighboring pixels will be correlated. The other plots are less smooth than an empirical covariance matrix from an isotropic Gaussian (left image of every pair), but tend to have no discernible structure.

Next, we give the covariances of CNN weights in layer 1 (Fig. A.4) and layer 2 (Fig. A.5). We have omitted layer 3 of the CNN because it is just a fully connected layer and also showed no interesting structure.

Finally, Fig. A.6 (left) measures the amount of covariance of every layer in the ResNet. We fit the lengthscale of a Gaussian distribution with squared exponential kernel, on the spatial correlations of the convolutional filters. The right-hand figure is the same as Fig. 2a.

### A.2  EMPIRICAL OFF-DIAGONAL COVARIANCES

We report results for the distributions of off-diagonal covariances for the respective second layers of our FCNN and CNN in Figure A.7. The empirical distribution of off-diagonal elements in the covariance matrices is shown as a histogram, overlaid with a kernel density estimate of the expected distribution if the weights were samples from an isotropic Gaussian. We see that the empirical covariance distributions are generally more heavy-tailed than the ideal ones, that is, the empirical weights generally have larger covariances than would be expected from isotropic Gaussian weights. Note that, as observed above, the strongest covariances by far are found spatially in the CNN weights, that is, between weights within the same CNN filter. We report the same results for the other layers in the following. The FCNN results are shown in Figures A.8 and A.9 and the CNN results in Figure A.10.

### A.3  THE INFLUENCE OF DATA AUGMENTATION ON THE COLD POSTERIOR EFFECT

When running the CIFAR-10 experiments with Bayesian ResNets with and without data augmentation, we find that data augmentation seems to significantly increase the cold posterior effect (Fig. A.11). Moreover, data augmentation seems to increase the performance of the models a lot at colder temperatures, but not at the true Bayes posterior $T = 1$. This suggests that data augmentation can also be one of the reasons for the cold posterior effect, as already hypothesized by Wenzel et al. (2020a) and Aitchison (2020b).

### A.4  SGD BASELINES

In terms of likelihood, calibration, and OOD detection, almost all our BNN models consistently outperformed the SGD baselines. The results including SGD are shown for FCNNs in Figure A.12, for CNNs in Figure A.13, and for ResNets in Figure A.14.

### A.5  ALTERNATIVE ACTIVATION FUNCTIONS

We repeated the experiments on MNIST with Bayesian FCNNs and CNNs and replaced the ReLU activation functions from Figure 4 and Figure 5 with sigmoid (see Fig. A.15) and tanh (see Fig. A.16) activations respectively. We observe that while the performances are overall worse than with ReLU activations (as is generally expected), the effects of the different priors are qualitatively very similar.

### A.6  HIGHER TEMPERATURES

In the main body of the paper, we followed Wenzel et al. (2020a) in showing only posteriors with temperatures $T \leq 1$, because we were interested in studying cold posteriors. Here, we also show results for warm posteriors, that is, $T > 1$. We see in Figure A.17 and Figure A.18 that these

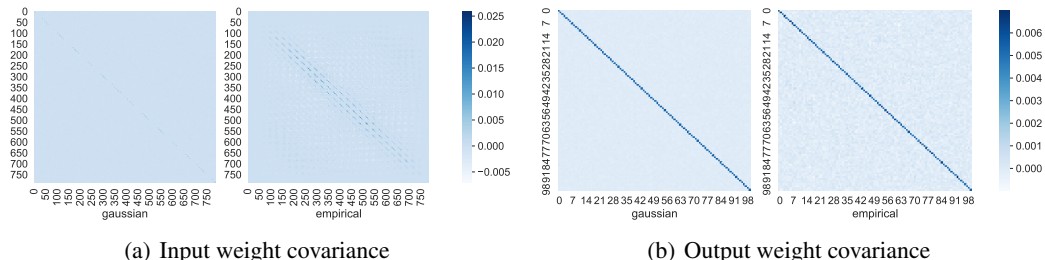

(a) Input weight covariance        (b) Output weight covariance

Figure A.1: FCNN layer 1 empirical covariances of the weights, trained with SGD on MNIST. We can see correlations in the spatial direction in the weights of the input layer (left). In the other directions, the covariance matrix is less smooth than we would expect from an isotropic Gaussian draw of the same size (left matrix of every pair), but otherwise has no discernible structure. This suggests that the weights are not isotropic Gaussian.

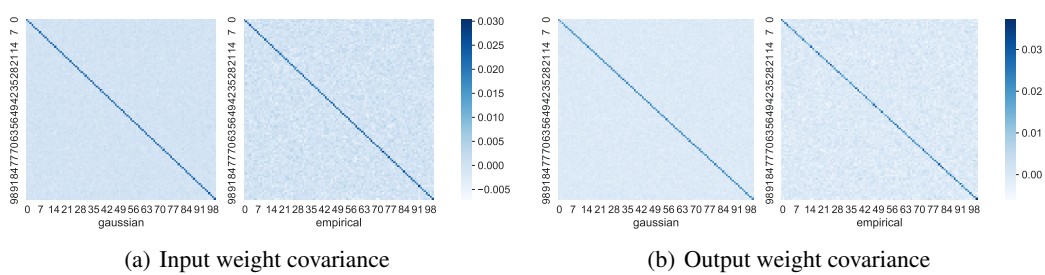

(a) Input weight covariance        (b) Output weight covariance

Figure A.2: FCNN layer 2 empirical covariances of the weights, trained with SGD on MNIST. The covariance matrix is less smooth than we would expect from an isotropic Gaussian draw, but has no discernible structure.

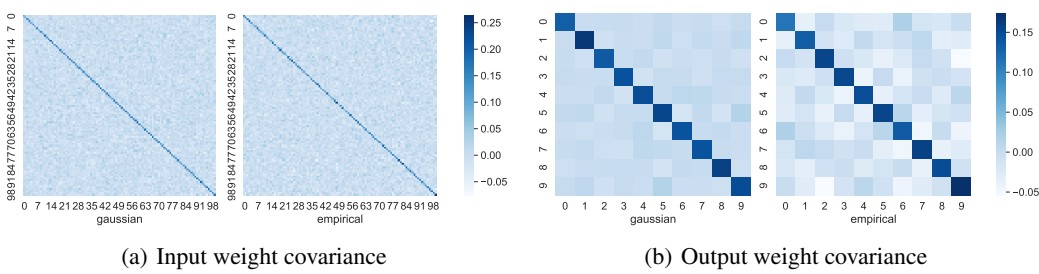

(a) Input weight covariance        (b) Output weight covariance

Figure A.3: FCNN layer 3 empirical covariances of the weights, trained with SGD on MNIST. The covariance matrix is less smooth than we would expect from an isotropic Gaussian draw, but has no discernible structure.

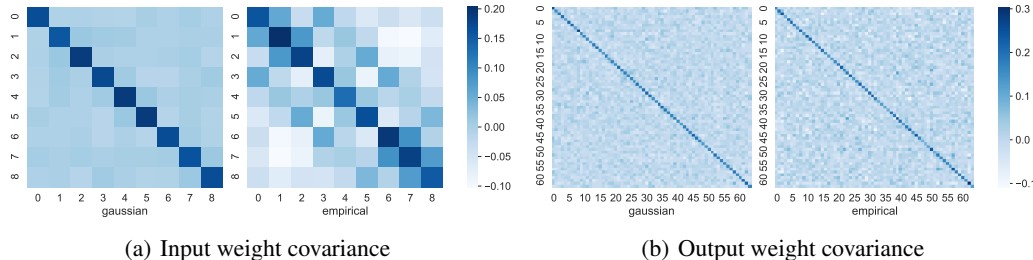

(a) Input weight covariance          (b) Output weight covariance

Figure A.4: CNN layer 1 empirical covariance of the weights, trained with SGD on MNIST. The input (also spatial) direction has correlations, also shown in Figure 2b. The output direction has no discernible structure.

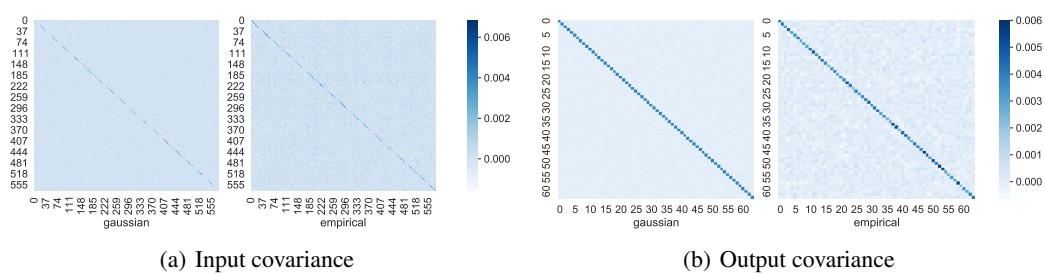

(a) Input covariance            (b) Output covariance

Figure A.5: CNN layer 2 empirical covariance of the weights, trained with SGD on MNIST. The input direction is less smooth than the isotropic Gaussian, and some low-rank structures can be observed. It should display the spatial correlation of Figure 2b. The output direction has no discernible structure.

warm posteriors generally do not improve the performance and that hence some of the priors (e.g., heavy-tailed priors in FCNNs) do indeed achieve their optimal performance for $T \approx 1$.

## A.7 DIFFERENT PRIOR VARIANCES

In the main text, we use models where the prior variance is chosen according to the He initialization (He et al., 2016), which is motivated by the conservation of the activation norm across the depth of the networks. Here, we see in Figures A.19, A.20, A.21, A.22, and A.23 that our main observations regarding the ordering of the different priors and the cold posterior effect still hold, even for different prior variances (in this case, four times larger and smaller than the He variance).

## A.8 DIFFERENT FCNN ARCHITECTURES

In the main text, we use FCNN models with three layers. Here, we see in Figure A.24 that our main observations regarding the ordering of the different priors and the cold posterior effect still hold, even for different architectures (in this case, between 2 and 4 layers).

## A.9 UCI REGRESSION

While the experiments in the main paper focus on image classification, we also performed BNN experiments on UCI regression tasks. The architecture is a 3-layer FCNN, the hidden layers are 64 units wide. We run GGMC for 30,000 epochs without minibatching on "boston", "energy", "yacht", and "wine", discarding runs where the potential diverges. For the other datasets, which are larger, we run 3000 epochs, also without minibatching. The learning rate is a flat $5 \cdot 10^{-5}$, and we do not use a cosine schedule.

Even with full batch MCMC, it is clear that the dynamics for regression networks are much less stable, especially at lower temperatures (which have a "sharper" potential landscape). Figure A.25

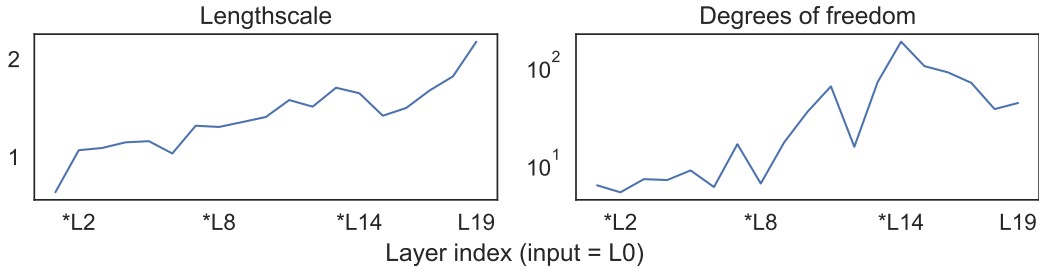

Figure A.6: Left: fitted lengthscale of a multivariate Gaussian with a squared exponential kernel (see eq. 4) to the data of Figure 3. All the entries of the SE covariance are positive, so this cannot capture all the features of the data, which has negative empirical covariance. Right: fitted degrees of freedom of a multivariate t-distribution, to same data. The empirical covariance was used in this case. The fitting criterion is the log-likelihood of the data. This is the same plot as Figure 2a.

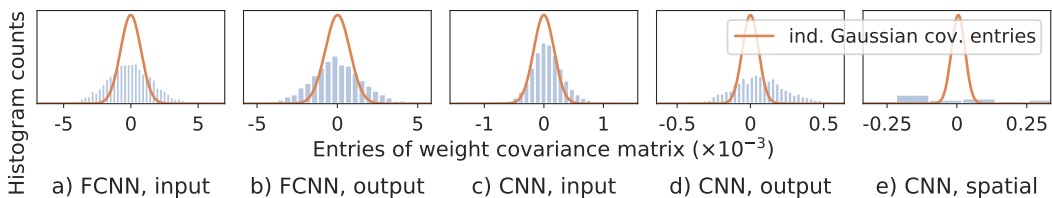

Figure A.7: Distributions of off-diagonal elements in the empirical covariances of the layer 2 weights of FCNNs and CNNs trained with SGD on MNIST. The empirical distributions are plotted as histograms, while the idealized random Gaussian weights are overlaid in orange. We see that the covariances of the empirical weights are more heavy-tailed than for the Gaussian weights.

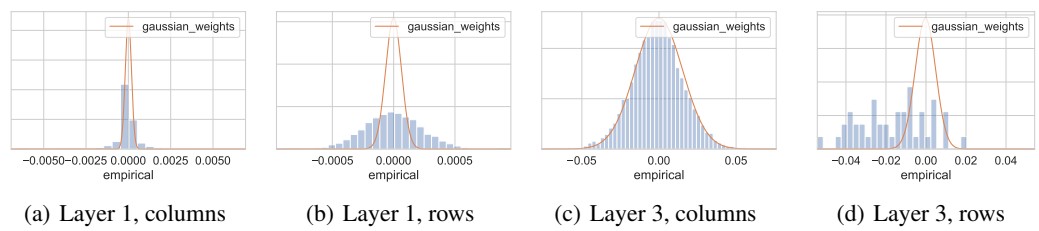

Figure A.8: Distributions of off-diagonal elements in the empirical covariances of the weights of the FCNN in layers 1 and 3. The empirical distributions are plotted as histograms, while the idealized random Gaussian weights are overlaid in orange. We see that the covariances of the empirical weights are more heavy-tailed than for the Gaussian weights.

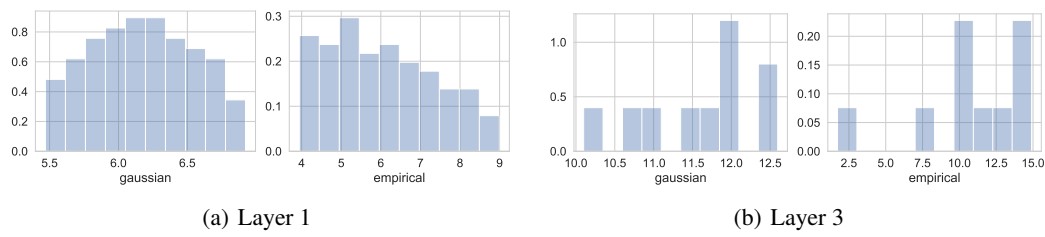

Figure A.9: Distributions of singular values of the weight matrices of the FCNN in layers 1 and 3. We see that the spectra of the empirical weights decay faster than the ones of the Gaussian weights.

shows that $T = 1$ is best for all datasets, in terms of the median mean squared error (MSE) as well as the quantiles and outliers. The priors are generally reasonably close in performance, such that it is harder in this case to strongly prescribe a certain prior choice. Of course, it is absolutely expect that different priors will be appropriate for different problems, especially when those problems are quite so distinct as regression and image classification.

The split-$\widehat{R}$ diagnostics are considerably higher for regression (Table A.1) than for the classification setting (Sec. A.10.2, for which the diagnostics look very good). Thus, the results here should be taken with a grain of salt. They are representative of how GGMC-trained BNNs behave at each of these temperatures and priors, but the results may be different for other (more accurate) ways of approximating the posterior. However, one thing is clear: UCI regression datasets exhibit no cold posterior effect, and for lower temperatures, the GGMC chains are less stable.

Table A.1: Diagnostics and median performance at temperature $T = 1$ for every prior and UCI dataset. The split-$\widehat{R}$ is generally high, which shows the chain has not fully explored the posterior. The best prior (in terms of median mean squared error, MSE) for each dataset is bolded, no prior is better overall. Additionally, each prior's performance is very similar for each dataset, which implies that the choice of prior does not matter much here (at least among these three).

|  | split-$\widehat{R}$ diagnostic | | | Median MSE | | |
|  | laplace | gaussian | student-t | laplace | gaussian | student-t |
|---|---|---|---|---|---|---|
| boston | 1.984856 | 1.664200 | 2.164047 | **0.051899** | 0.061199 | 0.056180 |
| concrete | 1.923906 | 1.960722 | 1.654736 | 0.962948 | 0.802203 | **0.728850** |
| energy | 2.026471 | 1.939554 | 2.309070 | 0.000759 | **0.000705** | 0.001500 |
| kin8nm | 1.523657 | 1.795404 | 1.609185 | 0.976947 | 1.520133 | **0.958203** |
| naval | 1.506857 | 1.700794 | 1.583324 | 1.241157 | **1.146679** | 1.679176 |
| power | 1.666368 | 1.738743 | 2.471113 | 0.156302 | 0.286625 | **0.148974** |
| protein | 1.574833 | 2.037037 | 1.510434 | **1.151049** | 1.296588 | 1.259395 |
| wine | 2.133220 | 1.936430 | 1.844044 | 0.674713 | 0.617795 | **0.604185** |
| yacht | 2.034536 | 1.880282 | 2.414279 | 0.000612 | **0.000559** | 0.000599 |

## A.10 INFERENCE DIAGNOSTICS

One of the main goals of our work is to make statements about the true BNN posteriors that are as accurate as possible. To this end, we closely monitored the accuracy of our inference algorithm. In order to check the correctness of our SG-MCMC inference, we estimated the temperature of the sampler using the two diagnostics from Wenzel et al. (2020a), namely the *kinetic temperature* and the *configurational temperature*.

The kinetic temperature is derived from the sampler's momentum $m \in \mathbb{R}^d$. The inner product $\frac{1}{d}m^\mathsf{T}M^{-1}m$, for the (in this case diagonal) mass matrix $M$, is an estimate of the scaled variance of the momenta. If the sampler is correct it should, in expectation, be equal to the desired temperature. The configurational temperature is slightly more involved and is discussed in Appendix A.10.1.

As an example, we show the estimated kinetic temperatures for our ResNet experiment on CIFAR-10 in Figure A.26. The desired temperature is shown as a dotted horizontal line. The kinetic temperatures for the other experiments look qualitatively similar and are shown in Appendix A.10.1. We see that the kinetic temperatures generally agree well with the true temperatures, so the sampler works as expected there. In contrast, the configurational temperature estimates can be somewhat larger than $T$, especially when $T$ is small (see Appendix A.10.1). This suggests that there could be small inference inaccuracies at low temperatures. However, these inaccuracies are small, and the configurational temperature certainly decreases as $T$ decreases, so there should be no impact on the overall trends.

We also computed the rank-normalized split-$\widehat{R}$ diagnostic Vehtari et al. (2021), which measures how well a collection of independent Markov chains have mixed. The split-$\widehat{R}$ is related to the ratio of between-chain and within-chain variances, and should be as close to 1 as possible. Given the complexity of neural network weight posteriors, we report the $\widehat{R}$ for the quantities we are interested in estimating (the y-values in Figs. 4 and 5). For every considered model and function, Table A.2

contains the worst (highest) $\widehat{R}$ estimate we obtained across all priors. Appendix A.10.2 contains a more detailed explanation and empirical $\widehat{R}$ estimates for different priors.

We can see that, for most experiments, the chains have mixed sufficiently. Only for the larger models (CIFAR10 ResNets)—and, to a lesser extent, Student-t FCNNs—the chains have mixed less well. Interestingly, for all convolutional networks, the correlated prior mixes best. This further supports its suitability as a prior for image data and CNNs.

### A.10.1 KINETIC AND CONFIGURATIONAL TEMPERATURE ESTIMATES

As described above, we use two temperature diagnostics (inspired by Wenzel et al. (2020a)): the *kinetic temperature* and the *configurational temperature*. The kinetic temperature is derived from the sampler's momentum $\boldsymbol{m} \in \mathbb{R}^d$. The inner product $\frac{1}{d}\boldsymbol{m}^{\mathsf{T}}\boldsymbol{M}^{-1}\boldsymbol{m}$, for the (in this case diagonal) mass matrix $\boldsymbol{M}$, is an estimate of the scaled variance of the momenta. It is always positive and should, in expectation, be equal to the desired temperature. In contrast, the configurational temperature is $\frac{1}{d}\boldsymbol{\theta}^{\mathsf{T}}\nabla H(\boldsymbol{\theta}, \boldsymbol{m})$, where $H(\boldsymbol{\theta}, \boldsymbol{m}) = -\log p(\boldsymbol{\theta} \mid \mathcal{D}) + \frac{1}{2}\boldsymbol{m}^{\mathsf{T}}\boldsymbol{M}^{-1}\boldsymbol{m} + \text{const}$ is the Hamiltonian. In expectation, this should also equal $T$. Unlike the kinetic temperature estimator, the configurational temperature estimator is not guaranteed to be always positive, even though the temperature *is* always positive. Using subsets of a parameter or momentum also yields estimators of the temperature.

In both cases, we estimate the mean and its standard error from a weighted average of parameters or momenta. That is, for each separate NN weight matrix or bias vector, we estimate its kinetic and configurational temperature using the expressions above. Then, we take their average and standard-deviation, weighted by the number of elements in that parameter matrix or vector.

We show the estimated temperatures of all our BNN experiments in Figures A.27, A.28, A.29, A.30, A.31, and A.32, as a mean $\pm$ one standard error. The desired temperature is shown as a dotted horizontal line. The kinetic temperatures generally agree well with the true temperatures, so our sampler works as expected there.

The configurational temperature estimates have a higher variance than the kinetic ones. Especially in the regime of small true temperatures, they often tend to slightly over- or underestimate the temperature. This is not surprising, since at low temperatures the noise in the gradients is dominated by the minibatching as opposed to the temperature noise. Correctly estimating the temperature from the gradients thus becomes harder.

Note that while the relative deviations can seem large in this regime, the absolute deviations are still quite small. Note also that while the conditioned momenta are strictly positive, the inner products between gradients and parameters can become negative in principle, which is why at low temperatures (close to 0) the configurational temperature estimates might sometimes be a bit below 0. Overall, the sampler is still within the tolerance levels of working correctly here, but there could be some small inaccuracies at low temperatures. However, judging from the shape of the actual tempering curves (see Sec. 4), the measures usually change more in the higher temperature regimes than in the lower ones, so there is no strong reason to believe that the inference at low temperatures was too inaccurate to support the results.

### A.10.2 BETWEEN-CHAIN AND WITHIN-CHAIN VARIANCES

The split-$\widehat{R}$ estimator measures the difference between posterior variance estimate in each chain, and between chains. It is roughly the square root of the between-chain variance divided by the within-chain variance (Vehtari et al., 2021, eq. 1–3). Its value is usually not smaller than 1, and a chain that has mixed well should have a value no larger than $\widehat{R} \leq 1.01$ (Vehtari et al., 2021). (Previously, a threshold of $1.1$ was considered enough (Gelman et al., 2013, Section 11.5).)

Neural network functional forms have a large number of parameter symmetries (for example, permutation invariance). Accordingly, the true BNN posterior should sample from all these modified parameters with probability proportional to their prior. However, for prediction purposes, it does not matter if the parameters are stuck in a single "permutation" and do not mix.

Therefore, for the purposes of this paper, we calculate the $\widehat{R}$ diagnostic not directly on the parameters, but on symmetry-invariant functions of the parameters. In practice, this amounts to evaluating the NN on a test set, and calculating the $\widehat{R}$ diagnostic for functions of the logits and the prior probability.

Table A.3: Estimated $\widehat{R}$ values for the different models and priors with respect to the loss.

|  | Gaussian | Laplace | Student-t | Correlated |
|---|---|---|---|---|
| MNIST FCNN | 1.000 | 1.001 | 1.006 | - |
| FashionMNIST FCNN | 1.000 | 1.000 | 1.007 | - |
| MNIST CNN | 1.000 | 1.000 | 1.002 | 1.000 |
| FashionMNIST CNN | 1.001 | 1.003 | 1.009 | 1.001 |
| CIFAR10 ResNet | 1.115 | 1.115 | 1.125 | 1.109 |
| CIFAR10 ResNet (augmented) | 1.057 | 1.054 | 1.047 | 1.066 |

Table A.4: Estimated $\widehat{R}$ values for the different models and priors with respect to the potential.

|  | Gaussian | Laplace | Student-t | Correlated |
|---|---|---|---|---|
| MNIST FCNN | 1.000 | 1.002 | 1.023 | - |
| FashionMNIST FCNN | 1.000 | 1.000 | 1.013 | - |
| MNIST CNN | 1.000 | 1.000 | 1.001 | 1.000 |
| FashionMNIST CNN | 1.000 | 1.002 | 1.007 | 1.000 |
| CIFAR10 ResNet | 1.166 | 1.147 | 1.171 | 1.139 |
| CIFAR10 ResNet (augmented) | 1.085 | 1.083 | 1.073 | 1.090 |

Tables A.3, A.4 and A.5 display the value of the diagnostic $\hat{R}$ for different such functions: the log-likelihood, the unnormalized log-posterior (potential), and the log-prior, respectively.

We employ the rank-normalized $\widehat{R}$ estimator (Vehtari et al., 2021, eq. 14) as implemented in the `ArviZ` library (Kumar et al., 2019).

The diagnostics are generally favorable ($\widehat{R} \leq 1.01$, mostly) for smaller NNs (FCNNs and 2-layer CNNs) and for MNIST. Within the ResNets applied to CIFAR10, the prior distribution with the $\widehat{R}$ closer to 1 is the correlated Gaussian. This provides evidence that inference is easier in the case of the correlated Gaussian, and therefore that the correlated Gaussian is a better prior (Gelman et al., 2013; Yang et al., 2015). This is because if the prior is good, the data are plausible simulations from it; so the posterior is close to the prior and will be easy to approximate.

## A.11 VARIATIONAL INFERENCE

In this paper, our experimental results have focused on inference with SG-MCMC, as we wished to obtain the most reliable posterior possible. However, non-sampling approaches such as variational inference (VI; e.g., Graves, 2011; Blundell et al., 2015; Dusenberry et al., 2020) and Laplace's method (e.g. Immer et al., 2021b) remain popular in the literature. Therefore, it might be valuable to understand the effect of the prior on the performance of these methods. In this section, we focus on variational inference (Wainwright et al., 2008), in particular the mean-field VI (MFVI) approach (Graves, 2011; Blundell et al., 2015).

Table A.5: Estimated $\widehat{R}$ values for the different models and priors with respect to the log prior.

|  | Gaussian | Laplace | Student-t | Correlated |
|---|---|---|---|---|
| MNIST FCNN | 1.000 | 1.005 | 1.101 | - |
| FashionMNIST FCNN | 1.000 | 1.003 | 1.104 | - |
| MNIST CNN | 1.001 | 1.002 | 1.013 | 1.001 |
| FashionMNIST CNN | 1.002 | 1.006 | 1.013 | 1.001 |
| CIFAR10 ResNet | 1.404 | 1.232 | 1.366 | 1.195 |
| CIFAR10 ResNet (augmented) | 1.274 | 1.346 | 1.264 | 1.198 |

Variational inference attempts to approximate the true intractable posterior $p(w|x, y)$ by a tractable approximate posterior $q(w)$ from an approximating family $\mathcal{Q}$ by maximizing the evidence lower bound (ELBO):

$$q^*(w) = \arg\max_{q \in \mathcal{Q}} \mathcal{L}(q; \lambda) = \arg\max_{q \in \mathcal{Q}} \mathbb{E}_q[\log p(y|x, w)] - \lambda \mathrm{KL}(q(w)||p(w)). \qquad (5)$$

For $\lambda = 1$, the ELBO is a true lower bound to the marginal likelihood of the model, and the true posterior is recovered as the optimal solution when $Q$ is the family of all distributions over $w$. For MFVI, we restrict the approximating distribution to be a fully-factorized Gaussian, that is, $q(w) = \prod_i \mathcal{N}(w_i|\mu_i, \sigma_i^2)$, so that there is no correlation structure in the approximate posterior. The variational parameters $\{\mu_i, \sigma_i\}$ can then be optimized using the reparameterization trick (Kingma & Welling, 2014; Rezende et al., 2014).

As with (SG-)MCMC, we can temper the posterior by adjusting $\lambda$, with $0 < \lambda < 1$ resulting in a "cold posterior". However, we note that apart from the case $\lambda = T = 1$, where we target the true posterior in both VI and MCMC, there is no straightforward, direct relationship between the cold posterior obtained in Eq. 1 and that obtained from Eq. 5 (for discussion see Wenzel et al. (2020a), particularly App. E).

### A.11.1 EXPERIMENTAL DETAILS AND RESULTS

We replicate the experiment in Sec. 4.3 for the ResNet architecture using CIFAR-10. We train each model for 1,000 epochs on batches of 500 augmented datapoints, using Adam (Kingma & Ba, 2015) with an initial learning rate of 0.01, which we reduce to 0.001 after 500 epochs. We are able to use these relatively high learning rates because we follow the parameterization introduced in Ober & Aitchison (2020); we also follow their step-wise tempering scheme for the first 100 epochs, which gradually increases the influence of the KL term. We use 1 sample from the approximate posterior for training and 10 samples for testing. Finally, we again run 5 replicates for each model.

We plot the results of this experiment in Figure A.33. We immediately make a few observations. First, the performance of MFVI is far worse than that of SG-MCMC on all metrics with the exception of calibration. We note that the performance at $\lambda = 1$ is particularly bad, which reflects the well-documented behavior that tempering with $\lambda < 1$ is required for decent performance with MFVI (e.g., Wenzel et al., 2020a). Finally, it does not seem that the choice of prior has much effect on the performance of MFVI, as all priors perform similarly. We hypothesize that this is largely due to the mean-field assumption imposed on the approximate posterior, which severely restricts its expressiveness and can lead to pathological behavior (Foong et al., 2019; Trippe & Turner, 2018). The mean-field assumption leads to a poor approximation to the true posterior, and therefore will not be as influenced by the choice of prior as SG-MCMC. However, we leave a full investigation of these effects to future work.

## B EVALUATION METRICS

When using BNNs, practitioners might care about different outcomes. In some applications, the predictive accuracy might be the only metric of interest, while in other applications calibrated uncertainty estimates could be crucial. We therefore use a range of different metrics in our experiments in order to highlight the respective strengths and weaknesses of different priors. Moreover, we compare the priors to the empirical weight distributions of conventionally trained networks.

### B.1 EMPIRICAL TEST PERFORMANCE

**Test error** The test error is probably the most widely used metric in supervised learning. It intuitively measures the performance of the model on a held-out test set and is often seen as an empirical approximation to the true generalization error. While it is often used for model selection, it comes with the risk of overfitting to the used test set (Bishop, 2006) and in the case of BNNs also fails to account for the predictive variance of the posterior.

**Test log-likelihood** The predictive log-likelihood also requires a test set for its evaluation, but it takes the predictive posterior variance into account. It can thus offer a built-in tradeoff between the

mean fit and the quality of the uncertainty estimates. Moreover, it is a proper scoring rule (Gneiting & Raftery, 2007).

## B.2 Uncertainty estimates

**Uncertainty calibration** Bayesian methods are often chosen for their superior uncertainty estimates, so many users of BNNs will not be satisfied with only fitting the posterior mean well. The calibration measures how well the uncertainty estimates of the model correlate with predictive performance. Intuitively, when the model is for instance 70 % certain about a prediction, this prediction should be correct with 70 % probability. Many deep learning models are not well calibrated, because they are often overconfident and assign too low uncertainties to their predictions (Ovadia et al., 2019; Wenzel et al., 2020b). When the models are supposed to be used in safety-critical scenarios, it is often crucial to be able to tell when they encounter an input that they are not certain about (Kendall & Gal, 2017). For these applications, metrics such as the expected calibration error (Naeini et al., 2015) might be the most important criteria.

**Out-of-distribution detection** The out-of-distribution (OOD) detection measures how well one can tell in-distribution and out-of-distribution examples apart based on the uncertainties. This is important when we believe that the model might be deployed under some degree of dataset shift. In this case, the model should be able to detect these OOD examples and be able to reject them, that is, refuse to make a prediction on them.

## C Details about the considered priors

We contrast the widely used isotropic Gaussian priors with heavy-tailed distributions, including the Laplace and Student-t distributions, and with correlated Gaussian priors. We chose these distributions based on our observations of the empirical weight distributions of SGD-trained networks (see Sec. 3) and for their ease of implementation and optimization. We now give a quick overview over these different distributions and their most salient properties.

**Gaussian.** The isotropic Gaussian distribution (Gauss, 1809) is the *de-facto* standard for BNN priors in recent work (e.g., Hernández-Lobato & Adams, 2015; Louizos & Welling, 2017; Dusenberry et al., 2020; Wenzel et al., 2020a; Neal, 1992; Zhang et al., 2019; Osawa et al., 2019; Immer et al., 2021b; Lee et al., 2017; Garriga-Alonso et al., 2019). Its probability density function (PDF) is

$$p(x; \mu, \sigma^2) = \frac{1}{\sqrt{2\pi\sigma^2}} \exp\left(-\frac{(x-\mu)^2}{2\sigma^2}\right)$$

with mean $\mu$ and standard deviation $\sigma$. It is attractive, because it is the central limit of all finite-variance distributions (Billingsley, 1961) and the maximum entropy distribution for a given mean and scale (Bishop, 2006). However, its tails are relatively light compared to some of the other distributions that we will consider.

**Laplace.** The Laplace distribution (Laplace, 1774) has heavier tails than the Gaussian and is discontinuous at $x = \mu$. Its PDF is

$$p(x; \mu, b) = \frac{1}{2b} \exp\left(-\frac{|x-\mu|}{b}\right)$$

with mean $\mu$ and scale $b$. It is often used in the context of (frequentist) *lasso* regression (Tibshirani, 1996).

**Student-t.** The Student-t distribution characterizes the mean of a finite number of samples from a Gaussian distribution (Student, 1908). Its PDF is

$$p(x; \mu, \nu) = \frac{\Gamma(\frac{\nu+1}{2})}{\Gamma(\frac{\nu}{2})\sqrt{\nu\pi}} \left(1 + \frac{(x-\mu)^2}{\nu}\right)^{-\frac{\nu+1}{2}},$$

where $\mu$ is the mean, $\Gamma$ is the gamma function, and $\nu$ are the degrees of freedom. The Student-t also arises as the marginal distribution over Gaussians with an inverse-Gamma prior over the variances

(Helmert, 1875; Lüroth, 1876). For $\nu \to \infty$, the Student-t distribution approaches the Gaussian. For any finite $\nu$ it has heavier tails than the Gaussian. Its $k$-th moment is only finite for $\nu > k$. The $\nu$ parameter thus offers a convenient way to adjust the heaviness of the tails. Note that it also controls the variance of the distribution, which is $\nu/(\nu - 2)$ (or else undefined). Unless otherwise stated, we set $\nu = 3$ in our experiments, such that the distribution has rather heavy tails, while still having a finite mean and variance.

**Multivariate Gaussian with Matérn covariance.**  For our correlated Bayesian CNN priors, we use multivariate Gaussian priors

$$p(\boldsymbol{x}; \boldsymbol{\mu}, \boldsymbol{\Sigma}) = \frac{1}{\sqrt{(2\pi)^d \det \boldsymbol{\Sigma}}} \exp\left(-\frac{1}{2}\|\boldsymbol{x} - \boldsymbol{\mu}\|_{\boldsymbol{\Sigma}^{-1}}^2\right)$$
$$\text{with} \quad \|\boldsymbol{x} - \boldsymbol{\mu}\|_{\boldsymbol{\Sigma}^{-1}}^2 = (\boldsymbol{x} - \boldsymbol{\mu})^\top \boldsymbol{\Sigma}^{-1} (\boldsymbol{x} - \boldsymbol{\mu}) \,,$$

where $d$ is the dimensionality.

In our experiments, we set $\boldsymbol{\mu} = \boldsymbol{0}$ and define the covariance $\boldsymbol{\Sigma}$ to be block-diagonal, such that the covariance between weights in different filters is 0 and between weights in the same filter is given by a Matérn kernel ($\nu = 1/2$) on the pixel distances, as applied by Garriga-Alonso & van der Wilk (2021) in the infinite-width case. Formally, for the weights $w_{i,j}$ and $w_{i',j'}$ in filters $i$ and $i'$ and for pixels $j$ and $j'$, the covariance is

$$\text{cov}(w_{i,j}, w_{i',j'}) = \begin{cases} \sigma^2 \exp\left(\frac{-d(j,j')}{\lambda}\right) & \text{if } i = i' \\ 0 & \text{else} \end{cases} , \tag{6}$$

where $d(\cdot, \cdot)$ is the Euclidean distance in pixel space and we set $\sigma = \lambda = 1$.

## D  IMPLEMENTATION DETAILS

**Training setup.**  For all the MNIST BNN experiments, we perform 60 cycles of SG-MCMC (Zhang et al., 2019) with 45 epochs each. We draw one sample each at the end of the respective last five epochs of each cycle. From these 300 samples, we discard the first 50 as a burn-in of the chain. Moreover, in each cycle, we only add Langevin noise in the last 15 epochs (similar to Zhang et al. (2019)). We start each cycle with a learning rate of $0.01$ and decay to 0 using a cosine schedule. We use a mini-batch size of 128.

For the SGD experiments yielding the empirical weight distributions, we use the same settings, but do not add any Langevin noise. We also do not use any cycles and just train the networks once to convergence, which in our case took 600 epochs.

We ran the experiments on GPUs of the type *NVIDIA GeForce GTX 1080 Ti* and *NVIDIA GeForce RTX 2080 Ti* on our local cluster. The main experiments (see Fig. 4 and Fig. 5) took around 10,000 GPU hours to run.

**FCNN architecture.**  For the FCNN experiments, we used a feedforward neural network with three layers, a hidden layer width of 100, and ReLU activations.

**CNN architecture.**  For the CNN experiments, we use a convolutional network with two convolutional layers and one fully connected layer. The hidden convolutional layers have 64 channels each and use $3 \times 3$ convolutions and ReLU activations. Each convolutional layer is followed by a $2 \times 2$ max-pooling layer.

**ResNet architecture and data augmentation.**  For the ResNet experiments on CIFAR-10, we use a ResNet20 architecture (He et al., 2016), equal to the one used in Wenzel et al. (2020a). For data augmentation, we pad all the images with 4 pixels on each border and then randomly crop out a 32x32 image out of that padded one and then randomly flip half of the images horizontally

**Software packages.** We implemented the inference and models with the PyTorch library (Paszke et al., 2019). To manage our experiments and schedule runs with several settings, we used Sacred (Greff et al., 2017) and Jug (Coelho, 2017) respectively. For the diagnostics, we also use Arviz (Kumar et al., 2019).

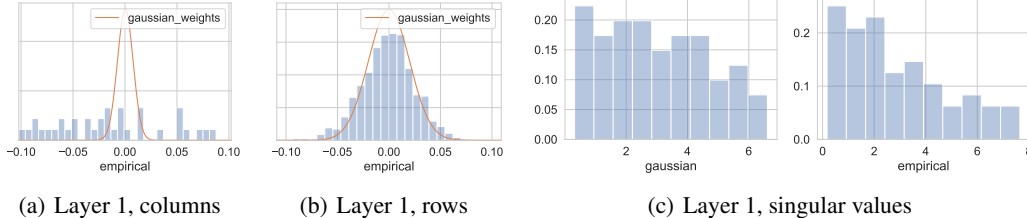

(a) Layer 1, columns  (b) Layer 1, rows  (c) Layer 1, singular values

Figure A.10: Distributions of off-diagonal elements in the empirical covariances of the weights and singular values of the CNN in the other layer. The empirical distributions are plotted as histograms, while the idealized random Gaussian weights are overlaid as an orange line. We see that the covariances of the empirical weights are more heavy-tailed than for the Gaussian weights and that the singular value spectrum for the empirical weights decays faster than the Gaussian ones.

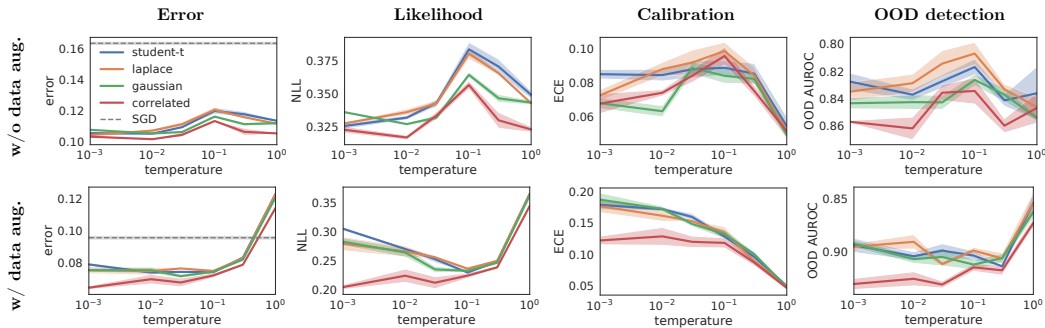

Figure A.11: Performances of Bayesian ResNets with different priors on CIFAR-10 with and without data augmentation in terms of different metrics. Data augmentation seems to increase the cold posterior effect.

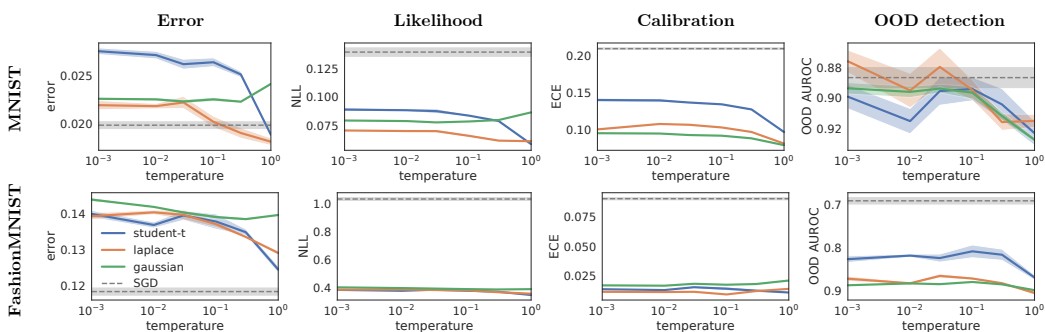

Figure A.12: Performances of fully connected BNNs with different priors on MNIST and Fashion-MNIST in terms of different metrics, compared to SGD solutions. The heavy-tailed priors perform better for Fashion MNIST, and perform better for MNIST at least for Laplace for error and NLL. heavy-tailed priors also eliminate the cold posterior effect (they get worse as temperature falls).

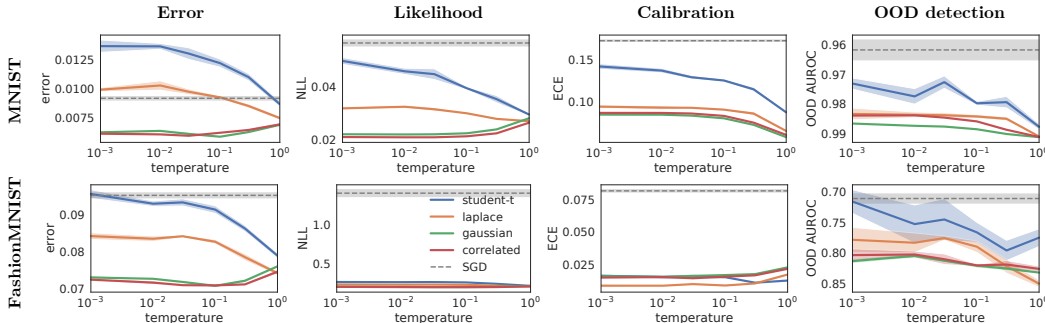

Figure A.13: Performances of convolutional BNNs with different priors on MNIST and Fashion-MNIST in terms of different metrics, compared to SGD solutions. The correlated prior generally performs better than the isotropic ones, but still exhibits a cold posterior effect, while the heavy-tailed priors reduce the cold posterior effect, but yield a worse performance.

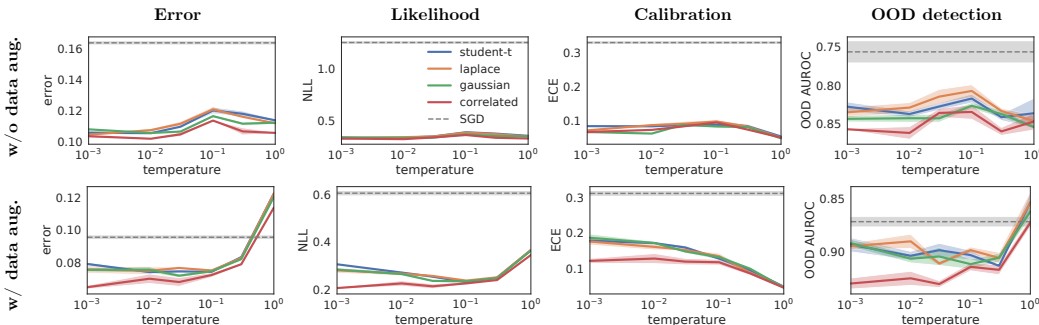

Figure A.14: Performances of Bayesian ResNets with different priors on CIFAR-10 with and without data augmentation in terms of different metrics, compared to SGD solutions. The correlated prior generally outperforms the other ones. Moreover, data augmentation seems to increase the cold posterior effect.

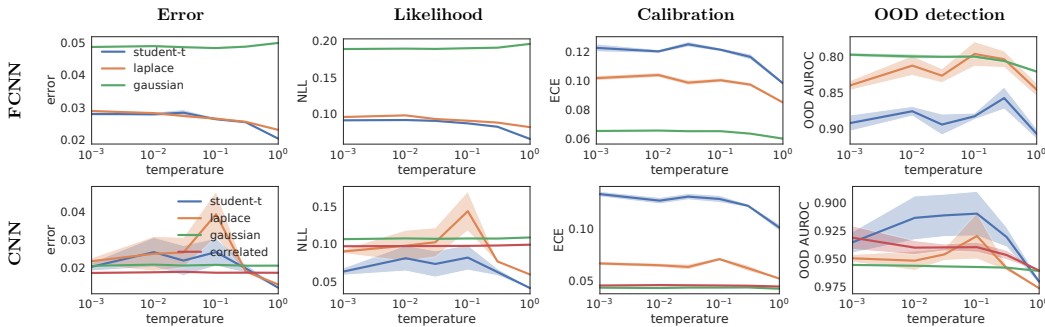

Figure A.15: Performances of fully connected and convolutional BNNs with sigmoid activation functions on MNIST. The observed effects are qualitatively similar to the ones with ReLU activations in the main body of the paper.

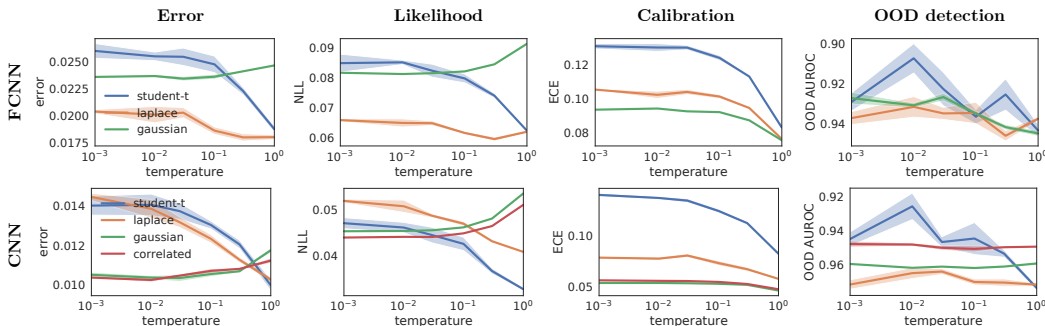

Figure A.16: Performances of fully connected and convolutional BNNs with tanh activation functions on MNIST. The observed effects are qualitatively similar to the ones with ReLU activations in the main body of the paper.

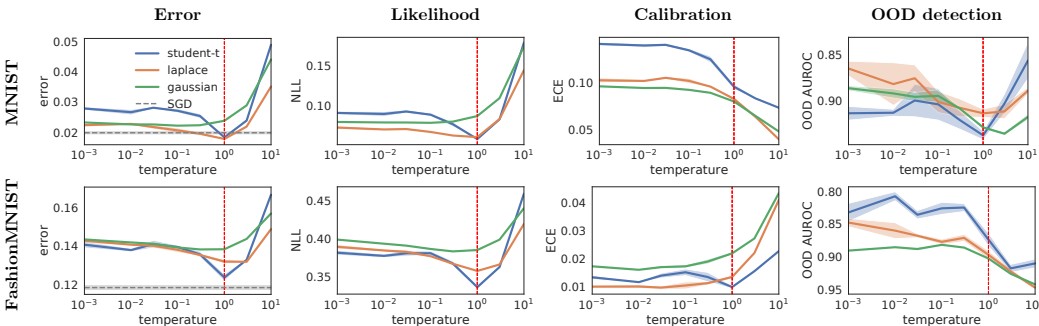

Figure A.17: Performances of Bayesian FCNNs with different priors on (Fashion-)MNIST, including temperatures $T > 1$. The performances generally do not improve for warm posteriors, such that $T \approx 1$ is indeed optimal for some priors.

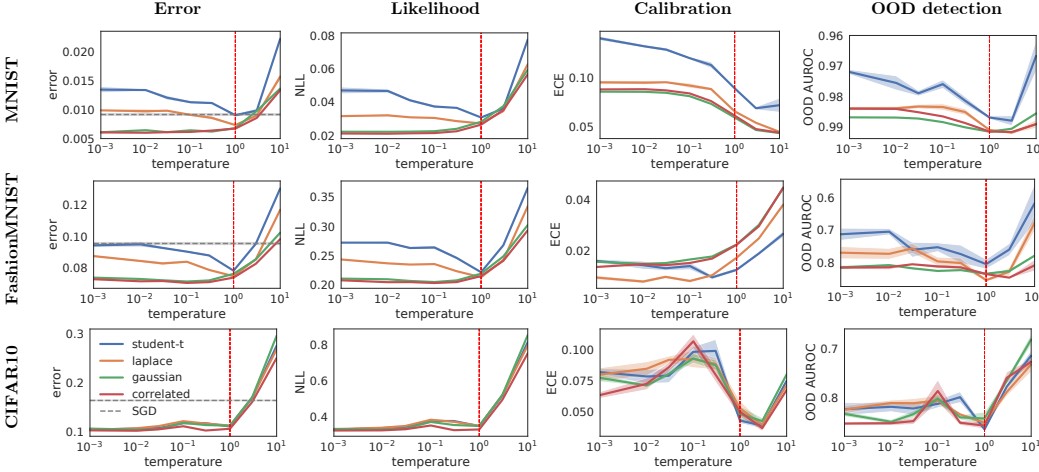

Figure A.18: Performances of Bayesian CNNs and Resnets with different priors on (Fashion-)MNIST and CIFAR, including temperatures $T > 1$. The performances generally do not improve for warm posteriors, such that $T \approx 1$ is indeed optimal for some priors. Note that here, we do not use data augmentation for CIFAR.

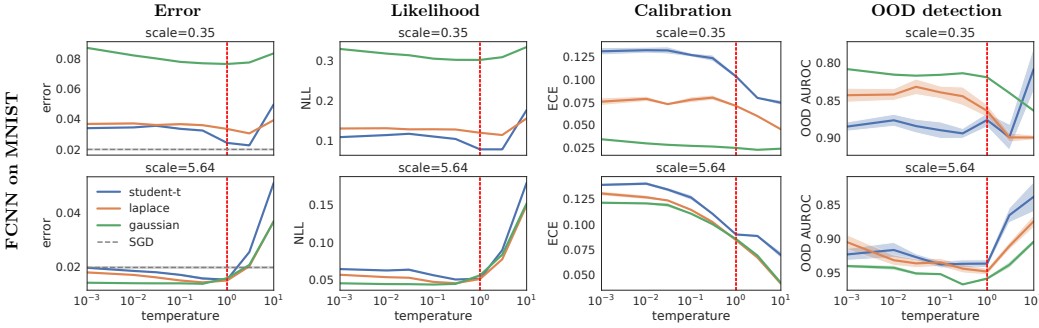

Figure A.19: Performances of Bayesian FCNNs with different priors and different prior variances on MNIST. The qualitative behavior is similar to the one for the He variance in the main text.

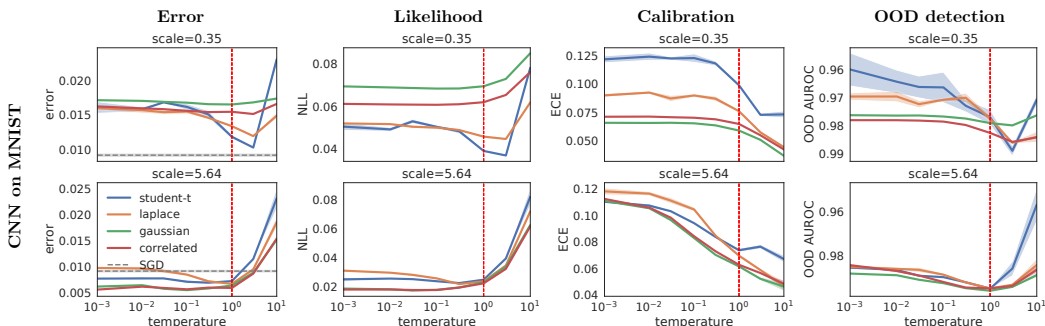

Figure A.20: Performances of Bayesian CNNs with different priors and different prior variances on MNIST. The qualitative behavior is similar to the one for the He variance in the main text.

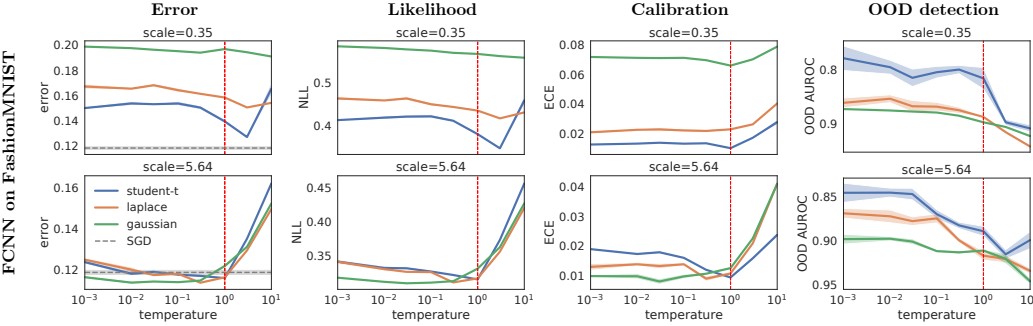

Figure A.21: Performances of Bayesian FCNNs with different priors and different prior variances on Fashion-MNIST. The qualitative behavior is similar to the one for the He variance in the main text.

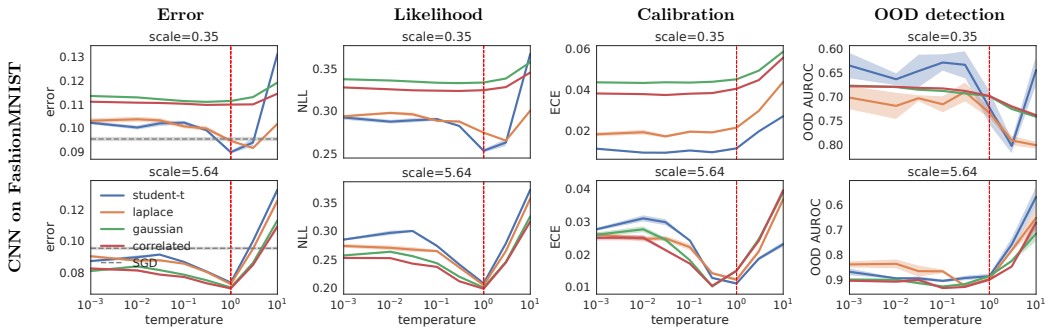

Figure A.22: Performances of Bayesian CNNs with different priors and different prior variances on Fashion-MNIST. The qualitative behavior is similar to the one for the He variance in the main text.

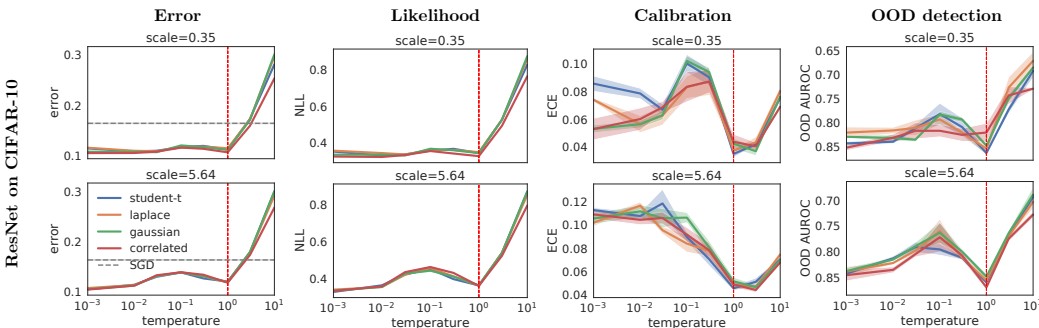

Figure A.23: Performances of Bayesian Resnets with different priors and different prior variances on CIFAR-10. The qualitative behavior is similar to the one for the He variance in the main text. Note that here, we do not use data augmentation.

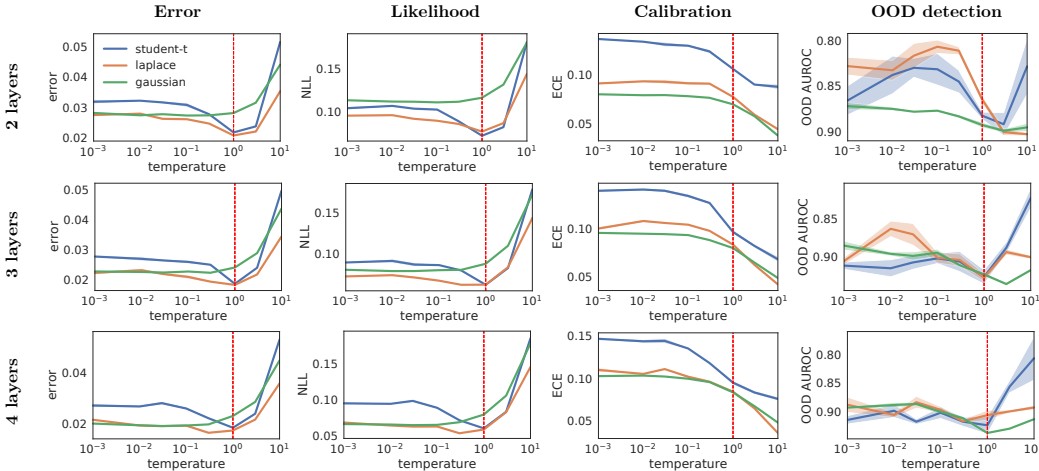

Figure A.24: Performances of Bayesian FCNNs with different priors and different depths on MNIST. The qualitative behavior for the different numbers of layers is similar to the one for three layers in the main text.

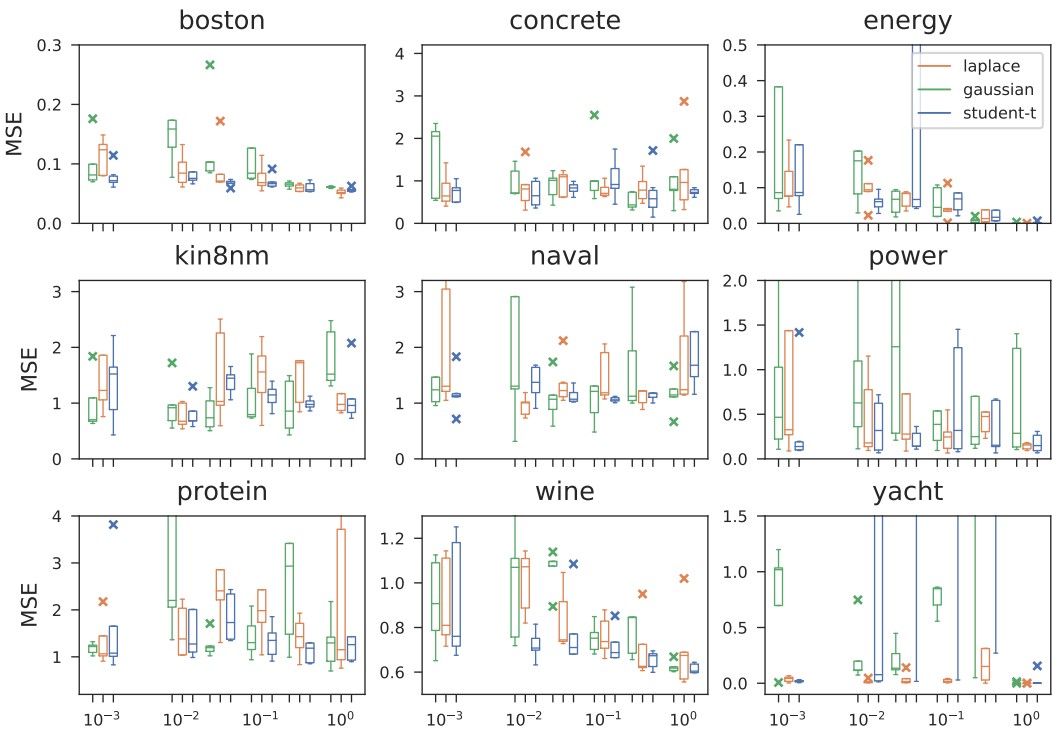

Figure A.25: Box plots of the mean-squared error of Bayesian FCNNs doing regression on UCI datasets. For each temperature, and prior, each box displays the median ±1.5 times the inter-quartile range. Outliers are plotted as ×. We exclude runs where the potential diverges. Temperature 1 is clearly best for all datasets, but otherwise there is no clear trend.

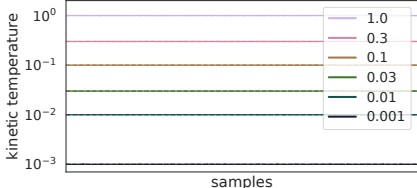

Figure A.26: Kinetic temperature diagnostics of the ResNet CIFAR-10 experiments with data augmentation. We see that the kinetic temperatures agree almost perfectly with the target temperature of the sampler.

Table A.2: Worst (highest) $\widehat{R}$ values for different models and neuron-permutation-invariant functions.

|  | Loss | Potential | Log-prior |
|---|---|---|---|
| FCNN MNIST | 1.006 | 1.023 | 1.101 |
| FCNN Fashion | 1.007 | 1.013 | 1.104 |
| CNN MNIST | 1.002 | 1.001 | 1.013 |
| CNN Fashion | 1.009 | 1.007 | 1.013 |
| ResNet CIFAR-10 | 1.125 | 1.171 | 1.404 |
| ResNet C.-10 (aug) | 1.066 | 1.090 | 1.346 |

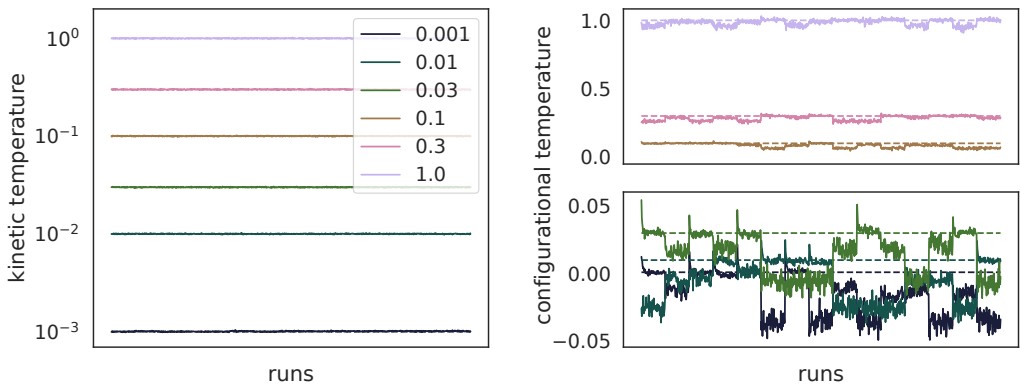

Figure A.27: Temperature diagnostics of the MNIST experiment with FCNNs.

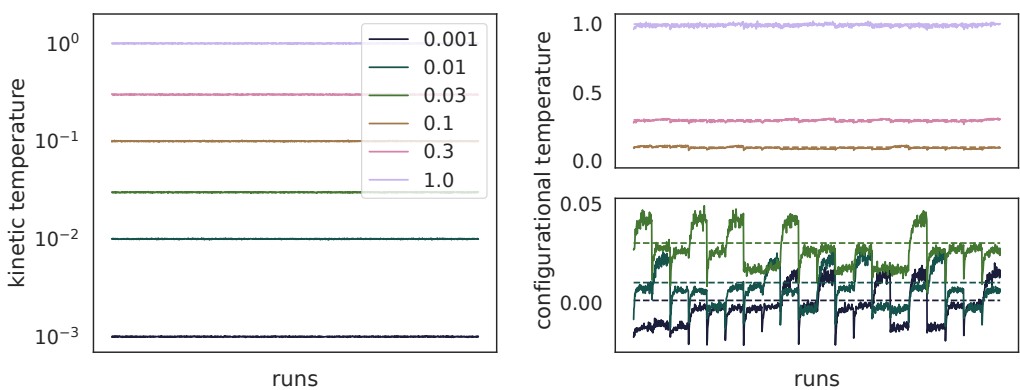

Figure A.28: Temperature diagnostics of the MNIST experiment with CNNs.

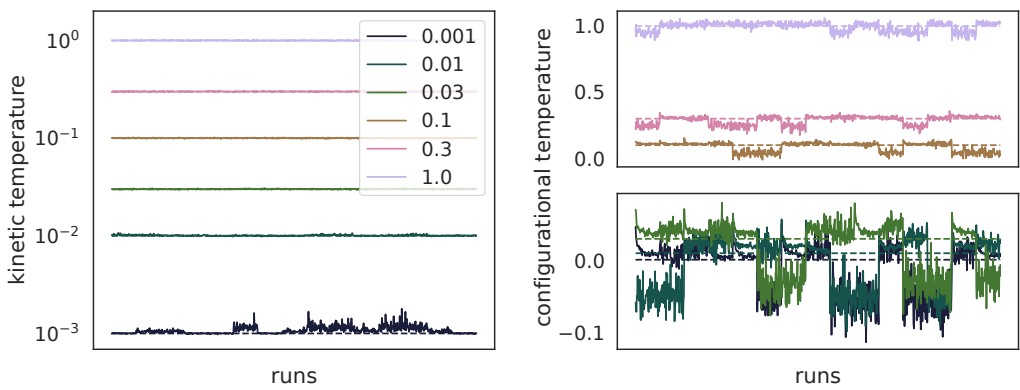

Figure A.29: Temperature diagnostics of the FashionMNIST experiment with FCNNs.

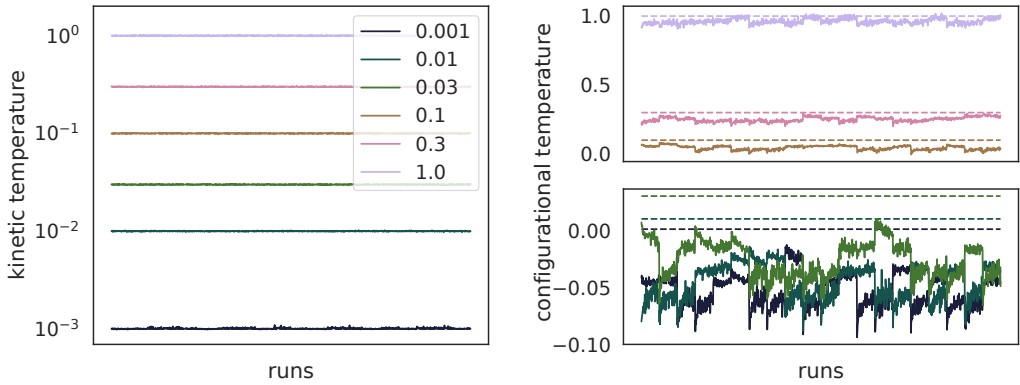

Figure A.30: Temperature diagnostics of the FashionMNIST experiment with CNNs.

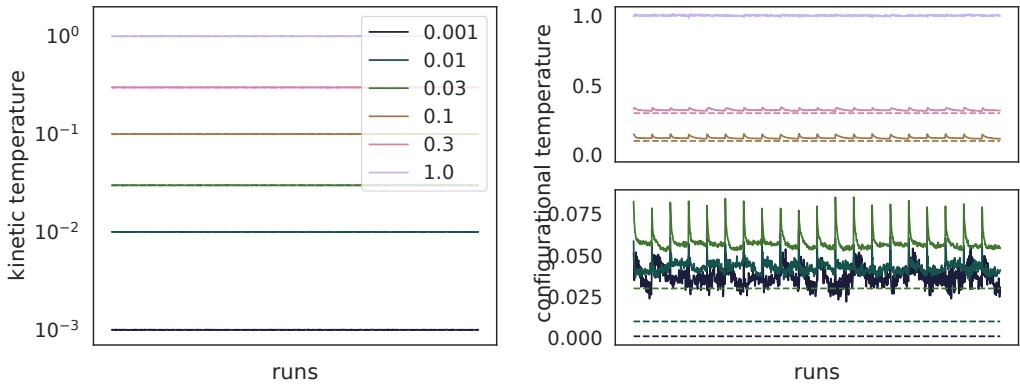

Figure A.31: Temperature diagnostics of the CIFAR-10 experiment with ResNets without data augmentation.

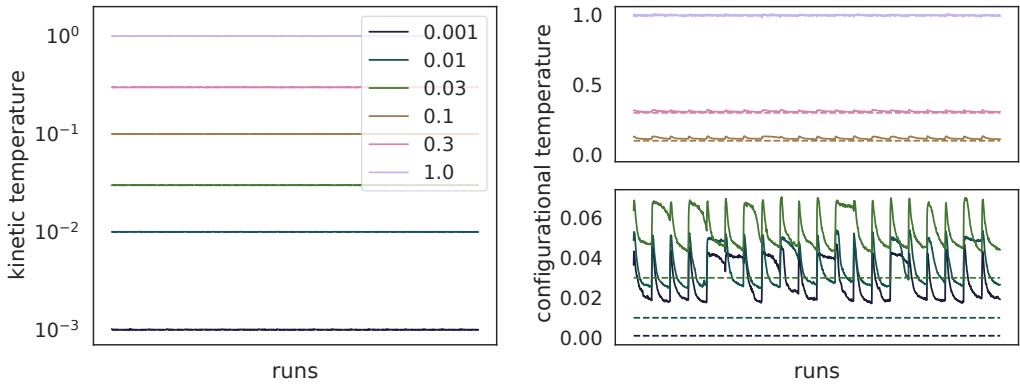

Figure A.32: Temperature diagnostics of the CIFAR-10 experiment with ResNets with data augmentation.

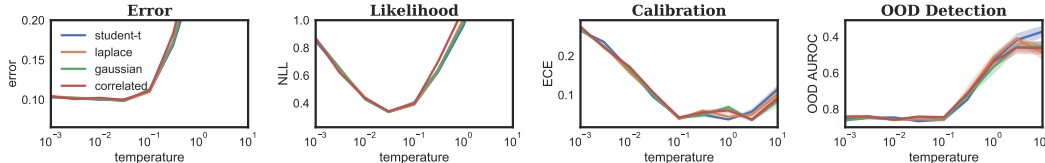

Figure A.33: Performances of mean-field variational inference ResNets with different priors on CIFAR-10. Note the reversed y-axis for OOD detection on the right to ensure that lower values are better in all plots. Shaded regions represent one standard error.

