# OpenReview forum: "Bayesian Neural Network Priors Revisited"
_ICLR.cc/2022/Conference — ICLR 2022 Poster_

### Official Review · Reviewer_yXaK · 2021-10-23

**Correctness:** 3
**Technical Novelty And Significance:** 3
**Empirical Novelty And Significance:** 3
**Recommendation:** 5
**Confidence:** 4

**Main Review:**

Strengths:
1) The paper took an empirical approach by first analyzing the weight distributions of the SGD trained model, then identifying the characteristics of these distributions, and finally proposing priors with the corresponding characteristics to facilitate the Bayesian learning. An expectation-maximization procedure is provided to justify the above approaches and empirical results support the choice of the priors.

2) The impact of these new priors on different type of networks are discussed, especially the cold-posterior effect which relates to the setting the the temperature in the posterior.

Weakness:
1) It would be great if the impact to the computational speed could be discussed further. Would the new priors drastically increase the computational demand, or the computation needed is similar to the isotropic gaussian prior?

2) As discussed in section 3 equation (2), SGD is used to sample from the posterior on w. However, it is unclear to me why SGD could be used to sample from the posterior weights of the networks. According to my understanding, a proper tool to sample from neural network posteriors is stochastic gradient Langevin dynamics (https://icml.cc/2011/papers/398_icmlpaper.pdf) rather than SGD. The "stochastic" part of SGD is with respect to the batch sampling from the data distribution, not with respect to the weight parameters. More clarification is needed regarding this choice of the weight sampler.

3) One motivation of the work from the introduction section is to impose proper prior distributions on the network weights, so that principled Bayesian inference can be carried out without temperature adjustment as in cold-posterior. However, according to my understanding, when the temperature is set to be 1, the results with the proposed priors do not always match or outperform the (non-Bayesian) SGD result (see Figure 4 and 5). This somewhat weakens the motivation and impact of the paper.

**Summary Of The Paper:**

This paper proposed to use distributions different from isotropic gaussian for Bayesian neural network priors. For fully connected neural networks the heavy-tailed distributions such as laplacian and student-t are used, while for convolutional neural networks multivariate gaussian with spatial correlations are employed. Empirical results show that these alternative priors give better results than isotropic gaussian on several image classification tasks. Cold-posteriors versions of the isotropic gaussian prior and the proposed priors are also provided for comprehensive comparison. Overall this is an interesting paper with novel discoveries.

**Summary Of The Review:**

This paper studied the important question of what would be a proper prior for Bayesian neural networks, and proposals were made based on empirical investigations of weight distributions of different neural networks. Experimental results on some image datasets validated the choice of the priors. In summary this is a nice paper with interesting findings. Still there are some technical details that need to be addressed (see my comments above).

---

> ### Author Response · Authors · 2021-11-14
> **Response**
>
> Thanks for your comments!
>
> ### Weakness 1
> Different priors have little to no impact on compute time, as we train for exactly the same number of steps etc.
> The only possible difference is in the computations of the gradient of the log-prior, but these are dominated by the actual convolution operations.
>
> ### Weakness 2
> We agree that SGD, as used in Sec. 3 doesn't do proper inference.  This, along with our motivation, is stated very explicitly in the main text:
> >Note that in the exploratory experiments here, we used SGD to perform MAP inference with a uniform prior (that is, maximum likelihood fitting).
> This avoids any prior assumptions obscuring interesting patterns in the inferred weights.
>
> And remember, we only do this to get inspiration for families of prior distribution to try in later, fully sampling-based experiments.
>
> ### Weakness 3
> In fact, in perhaps the most important case (correlated priors for ResNets; Fig 5 bottom), an improved prior actually increases the cold posterior effect. Importantly, this is a result in and of itself!
> The prior explanation of the CPE would suggest that priors that improve inductive biases and thus improve performance would reduce the cold posterior effect.
> We have established that this is not the case, by providing a prior (correlated Gaussians for ResNets) that improves performance but strengthens the cold posterior effect.
> This result has already influenced follow-up work:
>
> Nabarro S, Ganev S, Garriga-Alonso A, Fortuin V, van der Wilk M, Aitchison L. Data augmentation in Bayesian neural networks and the cold posterior effect. arXiv:2106.05586.
>
> We of course can't eliminate the possibility that an alternative prior exists that simultaneously improves performance and eliminates the cold posterior effect.  But we have found a prior that improves performance and strengthens to the cold posterior effect, which definitively establishes that not all priors that improve the inductive biases and performance also reduce the cold posterior effect.

---

### Official Review · Reviewer_sapj · 2021-10-31

**Correctness:** 4
**Technical Novelty And Significance:** 1
**Empirical Novelty And Significance:** 3
**Recommendation:** 8
**Confidence:** 4

**Main Review:**

*Update after rebuttal* I have read the authors’ response and other reviews. I would like to thank both the authors and other reviewers for providing an interesting discussion. Although the other reviews raise valid points on how to make the paper stronger, I believe it is worth publishing in its current form.

=======




Strong features:
1. Very well written paper. I really enjoy reading this paper. One can rarely see this style nowadays with more textbook, discussion-like language.
2. Very thorough experiments. Although it should be expected from a paper which main contribution is empirical evaluation of a phenomenon, but not always empirical papers hold this expectations. This submission does. Everything is considered from different angles, everything is tested - different activation functions, different variance, convergence, etc.
3. The paper is well placed in the context of the current literature.
4. The paper provides interesting insights into the default choice of the prior for BNNs as well as provides a recipe to explore this direction further.

Weak features:
I do not really see major weak points for this paper. Of course, there is always a room for improvement in anything, but in this case it would be "nice to have", rather that "should/must have". It would be interesting to see other types of priors, bigger datasets, other domains, but the current experiments have already taken 10,000 GPU hours as claimed by the authors.
The actual insight from the paper, i.e. that isotropic Gaussians are not the best choice of prior, may be not very exciting and surprising, but the paper provides a thorough analysis to show this and a recipe how to choose a better prior.

Even with a non-surprising result, I like this paper and enjoy reading it. I believe it provides useful and interesting discussion to the community and therefore I recommend acceptance of this paper. There is no anything particular novel in terms of the algorithms and methods in the paper, but rather a very well done empirical analysis that addresses a relevant problem.

Specific suggestions/comments (of different significance for assessment, but mostly for further improvement of the paper):
1. The acronym SGD in abstract is not introduced
2. The second paragraph in Introduction. "at lower temperatures" - a temperature has not been introduced yet
3. Eq. (1) - strictly speaking x and y should be defined separately
4. In the empirical analysis of weight distributions (Section 3), it is claimed that the posterior distribution reached by SGD would be a good choice for a prior. However, the posterior distribution family = the prior distribution family only when a prior is conjugate to a likelihood. This argument is therefore should be made more carefully
5. Figure 2a. It would be interesting to see some explanation/discussion of why the degree of freedom decreases between L14 and L19
6. Details on data augmentation is missing
7. Figure 3. It is claimed that the strength increases for later layers, but the max variance is less for them
8. The acronym SGLD is not introduced
9. Section 4. “To the best of our knowledge, this procedure constitutes the best SGLD-based inference approach” - before that it was stated that the authors used a combination of 3 different methods, therefore this claim looks unjustified
10. Figures 4-… What do shaded areas represent?
11. For the correlated Gaussian, it is not discussed why the Matern kernel in particular has been chosen
12. Sec A.1. “The covariances of the FCNN weights are shown in Figure A.5” – seems that it should be Figure A.1
13. Figure A.10 is not referred to in the text
14. Figure A.12 – not sure I can agree with the conclusions about heavy-tailed distributions made in the caption

**Summary Of The Paper:**

The paper provides an analysis of priors other than standard isotropic Gaussians for Bayesian neural network. The authors first empirically estimate the posterior distributions of the weights of a BNN and then use these distributions as a prior for these BNNs. They show that those priors allow better performance of BNNs. The paper discusses this analysis in the context of the cold posterior effect.

**Summary Of The Review:**

I vote for acceptance of this paper. I believe it provides a thorough empirical insights on the prior choice for BNNs and a recipe for future prior exploration.

---

> ### Author Response · Authors · 2021-11-14
> **Response**
>
> Thank you very much for your comments!
>
> 1.) The acronym SGD in abstract is not introduced
>
>   Fixed
>
> 2.) The second paragraph in Introduction. "at lower temperatures" - a temperature has not been introduced yet
>
>   Removed the earlier reference to "lower temperatures".
>
> 3.) Eq. (1) - strictly speaking x and y should be defined separately
>
>   Fixed.
>
> 4.) In the empirical analysis of weight distributions (Section 3), it is claimed that the posterior distribution reached by SGD would be a good choice for a prior. However, the posterior distribution family = the prior distribution family only when a prior is conjugate to a likelihood. This argument is therefore should be made more carefully
>
>   The classical EM algorithm does not require conjugacy.  Indeed, EM is just a strategy for performing variational inference, and variational inference with or without EM works well outside of the conjugate setting (see Neal RM, Hinton GE. A view of the EM algorithm that justifies incremental, sparse, and other variants. In Learning in graphical models. 1998)
>   And indeed, the goal is not to set the prior equal to the posterior for a particular dataset, as that would bake way too much information about that particular dataset into the prior.
>   Instead, the goal is to find the best prior within a family of simple parametric priors $p(w|\theta)$, such as independent Gaussian/Laplace/student-T.
>   Thus, the prior, $p(w|\theta)$, has nothing like the flexibility necessary to fully capture the posterior.
>   And indeed, the computation in Eq. 2: maximizing $\log p(w| \theta)$ for posterior samples of $w$ drawn from a different algorithm is straightforward for any non-conjugate model, $\log p(w| \theta)$, with a differentiable log probability.
>   Finally, $\theta$ would often just be e.g. the prior variance of a Gaussian prior.  But we could imagine a richer prior, including a categorical variable representing the type of prior (Gaussian/Laplace/student-T etc.)
>
> 5.) Figure 2a. It would be interesting to see some explanation/discussion of why the degree of freedom decreases between L14 and L19
>
>   The degrees of freedom is already pretty large here --- ranging from 30--100.  Distributions with degrees of freedom in this range are all _very_ close to Gaussian, so we are careful about overinterpreting this decrease.
>
> 6.) Details on data augmentation is missing
>
>   We pad all the images with 4 pixels on each border and then randomly crop out a 32x32 image out of that padded one and then randomly flip half of the images horizontally.  We have added this information to Appendix D.
>
> 7.) Figure 3. It is claimed that the strength increases for later layers, but the max variance is less for them
>
>   The strength *of spatial correlations* increases through layers.  But max-variance doesn't measure spatial correlations.  It is just the variance of the weights (taking the maximum over locations within the filter).  So is max-variance is actually entirely independent of the strength of spatial correlations.
>
> 8.) The acronym SGLD is not introduced
>
>   Fixed (by replacing SGLD -> SG-MCMC, which is defined).
>
> 9.) Section 4. “To the best of our knowledge, this procedure constitutes the best SGLD-based inference approach” - before that it was stated that the authors used a combination of 3 different methods, therefore this claim looks unjustified
>
>   We have deleted this claim.
>
> 10.) Figures 4-… What do shaded areas represent?
>
>   Added a note in the legend of Fig 4 and 5 to say that the shaded areas represent one standard error.
>
> 11.) For the correlated Gaussian, it is not discussed why the Matern kernel in particular has been chosen
>
>   We actually chose the form in Eq. 4 to very broadly capture the decay with distance of spatial correlations (Fig. 3), and this kernel turned out to be a special case of the Matern kernel.
>
> 12.) Sec A.1. "The covariances of the FCNN weights are shown in Figure A.5" – seems that it should be Figure A.1
>
> We have rewritten section A1 to ensure that these figures (A.1-A.6) are properly introduced in the text.
>
> 13.) Figure A.10 is not referred to in the text
>
> Good catch! We have update the paper to introduce this Figure (Fig. A.6) at the end of Section A.1, and noted that the right-hand side of it is already present as Fig. 2a.
>
> 14.) Figure A.12 – not sure I can agree with the conclusions about heavy-tailed distributions made in the caption
>
> Fixed the caption: "The heavy-tailed priors perform better for Fashion MNIST, and perform better for MNIST at least for Laplace for error and NLL. Heavy tailed priors also eliminate the cold posterior effect (they get worse as temperature falls)"

---

### Official Review · Reviewer_CSpQ · 2021-11-01

**Correctness:** 1
**Technical Novelty And Significance:** 2
**Empirical Novelty And Significance:** 2
**Recommendation:** 3
**Confidence:** 4

**Main Review:**

__Comments__:


- The results are kind of unstable, on the many plots (Figure 5, Figure A.19, Figure A.20) Gaussian prior behaves similarly or better than heavy-tailed analogs.


- The results on the cold-posterior effect are also unstable between models, which is reported in the paper.


- SGD baselines are undertrained (Figure 5, Figure A.11), e.g., ResNet-20 (w/ data aug.) error should be approx 8%, while 10% is reported. That is a quite important issue that may significantly flip the results of the experiments.


- The work studies only a single inference technique, while the results will not necessarily generalize between inference techniques. So, it is hard to make any general conclusions.

**Summary Of The Paper:**

The work studies prior distributions for Bayesian CNNs. The work report that conventionally used priors e.g., Gaussian poorly fit empirical distributions of the trined weights. The empirical distributions appear to be heavy-tailed and correlated. Thus the paper proposes to use "heavy-tailed priors" for FCNNs and correlated Gaussian priors for CNNs that tend to improve classification performance.

**Summary Of The Review:**

The work studies an interesting problem of prior selections for BNNs. At the same time, the results of the two main experiments are mixed and do not clearly demonstrate the importance of the heavy-tailed priors. The study also considers a single inference technique. Overall that makes me conclude that results are not generalizable and reliable.

---

> ### Author Response · Authors · 2021-11-14
> **Response**
>
> ### Consistency of cold posterior effects:
> While our analysis of the cold posterior effect doesn't overall make for a clean story, we would argue that this is significant in and of itself:
> The cold-posterior story would seem to suggest that all metrics get better as temperature falls.  We have shown that this isn't true: for instance, the behaviour of ECE in Fig. 4 depends strongly on the dataset and prior: increasing as temperature falls in some settings and decreasing in others.
> In addition, the prior explanation of the CPE would suggest that priors that improve inductive biases and thus improve performance would reduce the cold posterior effect.
> We have established that this is not the case, by providing a prior (correlated Gaussians for ResNets) that improves performance but strengthens the cold posterior effect.
> This result has already influenced follow-up work:
>
> Nabarro S, Ganev S, Garriga-Alonso A, Fortuin V, van der Wilk M, Aitchison L. Data augmentation in Bayesian neural networks and the cold posterior effect. arXiv:2106.05586.
>
> We of course can't eliminate the possibility that an alternative prior exists that simultaneously improves performance and eliminates the cold posterior effect.  But we have found a prior that improves performance and strengthens to the cold posterior effect, which definitively establishes that not all priors that improve the inductive biases and performance also reduce the cold posterior effect.
>
>
> ### SGD baseline
> We looked into this quite carefully, and couldn't find the difference to the Wenzel et al. 2020 paper causing the issue.  Nonetheless, the SGD results are just a sanity check.  They aren't material to the focus of the paper, which concerns whether different priors can improve performance or reduce the cold-posterior effect in _Bayesian_ neural networks.  We do not seek to make any claims about the relative performance of SGD and Bayesian neural networks.  Indeed, as in other work, we show that the performance of Bayesian CNNs/ResNets with T=1, corresponding to standard Bayesian inference is *worse* than SGD.
>
> Finally, it should be noted that in the Bayesian setting, notions of "undertraining" are much more complex than in standard deep learning.  It has frequently been observed by the authors that being more careful about Bayesian inference results in worse performance, and the cold posterior effect itself is a great example of such a phenomenon.
>
> ### Single inference method
> Ultimately, we are interested in the effect of different priors on the cold-posterior effect under *exact* inference.
> As that isn't possible, we use the best available proxy, SG-MCMC.
> This is widely regarded as the best approach, because it doesn't make unjustified assumptions that the posterior falls in some parametric family (such as Gaussian).
> Indeed, recent work:
>
> Izmailov P, Vikram S, Hoffman MD, Wilson AG. What Are Bayesian Neural Network Posteriors Really Like? ICML (2021)
>
> showed that SG-MCMC is a very good proxy for gold-standard HMC inference.
> Of course, it remains possible that other inference algorithms give very poor approximations to the true posterior, and therefore have very different behaviour with different priors.  But that wouldn't be a problem with our work.  It would be a problem with those "inference" algorithms which are doing such a bad job of estimating the true posterior, that they are unable to replicate the patterns we describe.

---

> > ### Comment · Reviewer_CSpQ · 2021-11-23
> > **Response**
> >
> > Dear authors,
> >
> > Thank you for your clarification and I extremely apologize for the late reply.
> >
> > TL;DR:
> > - My concerns remain true after the author's response.
> > - My main concern is that: _The results of the two main experiments are mixed and do not clearly demonstrate the importance of the heavy-tailed priors._
> > 	- e.g. The most important result "the most important case (correlated priors for ResNets; Fig 5 bottom)" is shown only on a single model-data pair.
> > 	- The pitch line in the paper is different from response one.
> >
> > - The concern seems to be supported by #R **Ergj**.
> > - We have a strong disagreement with #R **sapj**.
> > - I'm not sure that I get the score justification for #R **yXaK** correct.
> >
> > I read the full discussion. Comments to other revs:
> >
> > - **Ergj**:
> > 	- **Agree**: On the loosely supported claims. Specifically _"Experiments in Bayesian FCNNs showing that heavy-tailed priors give better classification performance than the widely-used Gaussian priors_" is not clearly demonstrated. Also, I would like to note that _"a better prior for ResNets_" results are obtained with potentially suboptimal HP settings.
> > 	- **Disagree**: On _"experiments lead to new (unanswered) questions_". I, unfortunately, cannot see it. For me the take-home message are i) weights are heavy-tailed/correlated ii) using these priors does not help mutch compared with other priors.
> > - **sapj**:
> > 	- **Agree** on Clarity
> > 	- **Disagree**:
> > 		- (Strongly) On _"Everything is considered from different angles, everything is tested - different activation functions, different variance, convergence, etc._".
> > 			- There is only one inference method tested.
> > 			- At the same time, the work draws general conclusions e.g., _"This provides evidence that the cold posterior effect arises due to a misspecification of the prior (Wenzel et al., 2020a) in FCNNs._".
> > 			- Main results (correlated Gaussians for ResNets) are consistent only for a single model-data pair.
> > 			- There is evidence (SGD baselines are undertrained) that HP tuning is not done carefully which might explain the difference between priors and mixed behavior.
> > 		- On _"The paper provides interesting insights into the default choice of the prior for BNNs_"
> > 			- I see the results are mixed and do not clearly demonstrate the importance of the heavy-tailed priors.
> > - **yXaK**
> > 	- Comment on _"One motivation of the work from the introduction section_"
> > 		- This is a bit complicated, Bayesian DNNs indeed do not always provide great performance.
> > 		- People hope to make great performance real one day. So it is more like a field state rather than a paper fault.
> > 		- Making your comment stronger, SGD solution is undertrained in the paper.
> > 		- Also paper is not testing new priors in the setting of small data, where prior influences the most.
> >
> >
> > Response:
> > - Consistency of cold posterior effects
> >
> > 	> _"We have shown that this isn't true: for instance, the behavior of ECE in Fig. 4 depends strongly on the dataset and prior: increasing as the temperature falls in some settings and decreasing in others._"
> >
> > 	This presentation line makes more sense to me. But that changes the gears compared to paper claims e.g., _"This provides evidence that the cold posterior effect arises due to a misspecification of the prior (Wenzel et al., 2020a) in FCNNs._".
> >
> > 	> We have established that this is not the case, by providing a prior (correlated Gaussians for ResNets) that improves performance but strengthens the cold posterior effect. This result has already influenced follow-up work.
> >
> > 	This is interesting! However, I'm still concerned about the sustainability of these results (suboptimal HPs as pointed out in initial rev, the demonstration only on the only a single pair of model-dataset).
> >
> > - SGD baseline
> >
> > 	- Unfortunately, I'm still concerned about it. This issue indicated that HP tuning might not be done carefully, which might explain the observed difference between priors.
> >
> > 	- I failed to find clarification on HP selection in the paper. Are all priors hold the same lr (+scheduler), wd, etc?
> >
> > - Single inference method
> >
> > 	- I agree that SG-MCMC can be considered as a strong inference method.
> > 	- But, I disagree that priors should be tested only with SG-MCMC inference.
> > 	- There is an option to test MAP inference. The other inferences e.g., VI with FF gaussian also can benefit from better priors.
> > 	- My concern: The presented general conclusions on the priors are misleading.

---

> > > ### Author Response · Authors · 2021-11-23
> > > **Quick response**
> > >
> > > Given the deadline is in about an hour, there isn't time for a full response.  But it is worth pointing out:
> > > 1. At very low temperatures, SGLD becomes equivalent to MAP.  So we are in essence already considering MAP inference (on the far-left of all temperature vs performance plots).
> > > 2. We have added depth 1 and 3 FCNNs (Fig. A24), which show similar benefits of heavy-tailed posteriors, so we now have multiple networks for this point.
> > > 3. In the Bayesian context, we need to set hyper parameters like learning rate so that there is a good match between the true posterior and SGLD approximate posterior. This is tough, but we have been very cautious and thorough in checking posterior convergence (Table A2-A5, Fig A26-32).  We have arguably been more careful than any past work on cold posteriors.  Importantly, this is the approach taken in all past work on cold posteriors.  For instance, the seminal and absurdly thorough work [1] again assesses accuracy of the posterior but doesn't explicitly optimize hyper parameters like learning rate.
> > > 4. We of course hold other hyperparmeters (lr, wd etc) the same for different priors. Frankly, we could probably have got cleaner results if we'd fiddled the hyperparameters!
> > >
> > > [1] Wenzel F, Roth K, Veeling BS, Świątkowski J, Tran L, Mandt S, Snoek J, Salimans T, Jenatton R, Nowozin S. How good is the bayes posterior in deep neural networks really?. arXiv preprint arXiv:2002.02405. 2020 Feb 6.

---

### Official Review · Reviewer_Ergj · 2021-11-03

**Correctness:** 2
**Technical Novelty And Significance:** 2
**Empirical Novelty And Significance:** 4
**Recommendation:** 6
**Confidence:** 5

**Main Review:**

# Weaknesses

 * There is no clear take-at-home message.
 * The authors generalize their results on *one* FCNN to *all* FCNNs and on *one* CNN to *all* CNNs, on one task. The consistency across models and datasets should have been checked.
 * There is no heuristics or explanation of the observed results (except for the influence of the data augmentation on the "cold posterior effect".

# Strengths

The experimental report is interesting for researchers who want to have a better intuition of the influence of the prior.

# EDIT

I have taken note of the modification made by the authors in the Discussion section, which moderates a bit the main claims. But I do not change my recommendation, since the main claims (in the first sections) are still too affirmative regarding the experiments.

Anyway, as an impressive empirical work, this paper deserves to be published.

**Summary Of The Paper:**

This paper presents numerous empirical facts about the distribution of the weights after training a neural network. The authors considered one FCNN, one CNN, and one ResNet, and computed several statistics on their weights.

The authors also propose, in a Bayesian setup, to compare the performance of trained BNNs with different priors.

**Summary Of The Review:**

Weak accept.

Despite the numerous and contradictory experiments (heavy tailed priors are better for FCNNs, but worse for CNNs), and the difficulty the authors have to interpret their results, the empirical results are significant enough for helping other researchers to guide their research.

For instance:
 * a heavy-tailed distribution of the weights is not necessarily a problem;
 * the distribution of weights after training a FCNN and a CNN may be structurally different;
 * the "cold posterior effect" can be partly explained by data augmentation;
 * the "cold posterior effect" can be removed when using a prior with heavier tail.

A major issue remains: the authors should have checked that their results about "FCNNs" and "CNNs" are consistent for *several* FCNNs/CNNs and several datasets.

---

> ### Author Response · Authors · 2021-11-14
> **Response**
>
> Thanks for your positive comments!
>
> As noted by Reviewer 3 (sapj), our experiments are already _very_ extensive.
> > Very thorough experiments. Although it should be expected from a paper which main contribution is empirical evaluation of a phenomenon, but not always empirical papers hold this expectations. This submission does. Everything is considered from different angles, everything is tested - different activation functions, different variance, convergence, etc.
>
> In particular, we consider:
> * 4 datasets: MNIST, FashionMNIST, CIFAR-10, UCI (Appendix Table A1).
> * 2 FCNNs: one for image classification (classifier output; 100 units in hidden layers) and one for UCI regression (Appendix A.8; regression output; 64 units in the hidden layers)
> * 2 CNNs: A shallow CNN (two convolutional and one fully connected layer; the hidden convolutional layers have 64 channels each and use $3\times 3$ convolutions. Each convolutional layer is followed by a $2\times 2$ max-pooling layer).  And a ResNet-20 (see Sec. 4.3 and Appendix D for details).
> * Networks with/without data augmentation (Fig. A11,A14).
> * Networks with different activation functions (sigmoid Fig. A15; tanh Fig. A16)
> * Networks with different prior variances (Fig. A19-22).
> * Networks with different priors over weights.
>
> While our analysis of the cold posterior effect doesn't overall make for a clean story, we would argue that this is significant in and of itself:
> The cold-posterior story would seem to suggest that all metrics get better as temperature falls.
> We have shown that this isn't true: for instance, the behaviour of ECE in Fig. 4 depends strongly on the dataset and prior: increasing as temperature falls in some settings and decreasing in others.
> In addition, we have found a better prior for ResNets (the correlated prior on weights in Fig. 5 bottom) which actually increases the magnitude of the CPE.  This gives evidence against the idea that the CPE is primarily caused by using the wrong prior: if it was, then we'd expect an improved prior to reduce (not increase) the magnitude of the CPE.
> This result has already influenced follow-up work:
>
> Nabarro S, Ganev S, Garriga-Alonso A, Fortuin V, van der Wilk M, Aitchison L. Data augmentation in Bayesian neural networks and the cold posterior effect. arXiv:2106.05586.
>
> Of course we can't eliminate the possibility that an alternative prior exists that simultaneously improves performance and eliminates the cold posterior effect. But we have found a prior that improves performance and strengthens to the cold posterior effect, which definitively establishes that not all priors that improve the inductive bias and performance also reduce the cold posterior effect.

---

> > ### Comment · Reviewer_Ergj · 2021-11-16
> > **Claims still loosely supported**
> >
> > I maintain that the claims about FCNNs and CNNs are not well supported. The example given in Section A.8 is not in favor of the authors: choosing heavy-tailed priors does not improve the results on a FCNN in a particular regression setting. So, I consider the claim "Experiments in Bayesian FCNNs showing that heavy-tailed priors give better classification performance than the widely-used Gaussian priors" as misleading and very speculative.
> >
> > Actually, there is only *one* architecture and type of dataset to support this claim. The only other experiment run with another FCNN and a very different dataset leads to an unclear result (see Section A.8). Even worse, the formulation of the claim is doubtfully specific: "[...] heavy-tailed priors give better *classification* performance than [...]". The reader may interpret the claim as "cherry-picked", since it is true in only *one* *classification* task, but untrue in one *regression* task. Synthetic datasets and FCNNs with 1, 2, 4 layers may have been tested with computation time similar to the existing experiments.
> >
> > Regarding this claim, the variety of experiments is insufficient, despite the large number of experiments made by the authors. This holds also for CNNs (ResNets cannot be interpreted as CNNs in this setting: these two kinds of architecture are structurally different, do not evolve in the same way, the proofs of convergence are very different...).
> >
> > Overall, I vote to accept this paper, since the numerous and various experiments lead to new (unanswered) questions. Checking every claim thoroughly may be done in future works, but, for this paper, I would recommend to write milder claims and possibly a plan for future experiments (in order to check the current claims).

---

> > > ### Author Response · Authors · 2021-11-20
> > > **Response**
> > >
> > > Thanks again for you comments.
> > >
> > > We have added a new Fig. A24 which shows the effects of different priors on FCNNs of depths 2, 3 and 4.  In all architectures, the heavy tailed Laplace prior gives improved predictive performance and largely eliminates the cold posterior effect.  This is consistent with our original results in networks with depth 3.
> > >
> > > We have updated the Discussion to emphasise the UCI results:
> > > > Importantly though, we do not expect there to be one "universal" prior that improves performance in all architectures and all tasks.
> > > The best prior is almost certain to be highly task and architecture dependent, and indeed we found that heavy tailed priors offer little or no benefits for regression on UCI datasets (Sec. A9).
> > >
> > > In addition, the abstract makes quite a specific claim, restricting us not only to classification, but even more narrowly to image classification: "We show that building these observations into priors can lead to improved performance on a variety of image classification datasets."
> > >
> > > We have updated the text to distinguish ResNets and CNNs more carefully.  In the previous response, we had only meant to claim that we had considered two network architectures with convolutional layers.

---

### Decision · Program_Chairs · 2022-01-20

**Decision:**

Accept (Poster)

**Comment:**

This paper provides some empirical investigation of the choice of the prior distribution for the weights in Bayesian neural networks. It shows empirically that, when trained via SGD, weights in feedforward neural networks exhibit heavy-tails, while weights in convolutional neural networks are spatially correlated. From this observation they show that the use of such priors leads to some improved performances compared to the iid Gaussian prior in some experimental settings.

Reviewers have conflicting views on this paper, that have not been reconcilied after the author's response and the discussion.
On the plus side, the paper is very well written, the experimental part is carefully conducted, and provides some insights on the choice of the prior in Bayesian neural networks, which could lead to further developments.
On the negative side, the claims made in the introduction are not fully supported by the experiments (the claims have been slightly amended in the revised version), and the take-home message is not so clear. In particular, Bayesian approaches with the proposed priors still underperform compared to SGD without tempering. The authors could also have considered a broader sets of experiments.

Overall, I think the contributions outweight the limitations of this paper, and I would recommend acceptance.